



# Geodiversity primarily shapes large-scale limnology and aquatic species distribution in the northern Neotropics

Laura Macario-González[1,2], Sergio Cohuo[2,3], Philipp Hoelzmann[4], Liseth Pérez[2], Manuel Elías-Gutiérrez[5], Margarita Caballero[6], Alexis Oliva[7], Margarita Palmieri[8], María Renée Álvarez[8] and Antje Schwalb[2]

[1]Tecnológico Nacional de México/I. T. de la Zona Maya. Carretera Chetumal-Escárcega Km 21.5, ejido Juan Sarabia, 77965 Quintana Roo, México.
[2]Institut für Geosysteme und Bioindikation, Technische Universität Braunschweig, Langer Kamp 19c, 38106 Braunschweig, Germany.
[3]Tecnológico Nacional de México/I. T. de Chetumal. Av. Insurgentes 330, Chetumal, 77013 Quintana Roo, México.
[4]Institut für Geographische Wissenschaften, Freie Universität Berlin, Malteserstrasse 74-100, 12249 Berlin, Germany.
[5]El Colegio de la Frontera Sur (ECOSUR), Unidad Chetumal, Av. Centenario km 5.5, Chetumal, 77014 Quintana Roo, México.
[6]Instituto de Geofísica, Universidad Nacional Autónoma de México, Ciudad Universitaria, 04510 Ciudad de México, México
[7]Asociación de municipios del lago de Yojoa y su Área de influencia (Amuprolago), Aldea Monte Verde, Santa Cruz de Yojoa, Cortés, 21000 Honduras.
[8]Departamento de Biología. Universidad del Valle de Guatemala, 01015 Guatemala, Guatemala.

*Correspondence to*: Sergio Cohuo (sergio.cd@chetumal.tecnm.mx)

**Abstract.** Geodiversity is recognized as one of the most important drivers of ecosystems characteristics and biodiversity globally. However, in the northern Neotropics, the contribution of highly diverse landscapes and geological history in structuring large-scale patterns of limnology and aquatic species associations remain poorly understood. We evaluate the interaction between geodiversity, limnology and freshwater ostracode assemblages to explore drivers of aquatic ecosystem structure from southern Mexico to Nicaragua. Cluster analysis based on geological, limnological, sedimentological and mineralogical characteristics of 76 aquatic ecosystems (karst, volcanic, tectonic) reveal two main limnological regions: (1) Karst terraces from the Yucatán Peninsula and northern Guatemala, and (2) volcanic terrains of the Guatemalan highlands, El Salvador and Honduras mid-elevations, and Nicaraguan lowlands. In addition, seven limnological subregions were recognized, attesting high limnological heterogeneity. Principal Component Analysis identifies water ionic and sediment composition as most influential for aquatic ecosystem classification and given their source of formation, we recognized geology as the fundamental determinant for large-scale limnological patterns. Multiparametric analyses based on biological data revealed that species association represent disjunct faunas. For instance, five species associations are closely related to limnological regions. Structural equation modelling reveals a strong influence of limnology and elevation on the biological composition, but geodiversity resulted in the most important driver. The confinement of species associations is attributed to regional lake geochemistry. We deduce a linear, continuous and primary influence of geodiversity for limnological structure, geochemical properties and aquatic biological composition in Lakes of the northern Neotropical region.



## 1 Introduction

Geodiversity is defined as the natural variety of geological, geomorphological and hydrological features of a given landscape (Gray, 2019). Geodiversity is responsible for aspects of regional or local climate, hydrology and soil properties (Vartanyan,

2006a; Hu et al., 2020). At interaction with biosphere and atmosphere, geodiversity can also contribute, via sedimentary delivery to the input of nutrients into ecosystems, and modify the chemical composition of environments (Vartanyan, 2006b; Bravo-Cuevas et al., 2021). All these features have been fundamental for ecosystem development and biological evolution at scales ranging from local to global (Mittelbach et al., 2007; Etienne and Apol, 2009; Smith et al., 2010; Bryson et al., 2013; Gillespie and Roderick, 2014; Steinbauer et al., 2016).

Biodiversity is dynamic and evolutionary processes and changes in species composition and distribution may take place at faster rates or even in absence of geological variability. This implies that geodiversity may have different degrees of contribution in shaping biodiversity at regional scales. In areas with highly dynamic and complex biological systems such as the Tropics (Rull, 2011; Antonelli et al., 2018; Matzke-Karasz et al., 2019; Moguel et al., 2021), it is therefore difficult to understand the contribution of geodiversity in shaping the distribution of past and modern biota (Rossetti and Toledo, 2016).

Exploring the role of geodiversity for the structure of landscape and biological systems is the key to understand evolutionary traits and mechanisms controlling species distribution, with further application to conservation, ecosystem management and to predict ecosystem responses under changing climate scenarios (Martiny et al., 2006; Hulsey et al., 2010; Jiménez-Alfaro et al., 2018).

The northern Neotropics extend from central-southern Mexico to Central America, including the Caribbean, and it is

characterized by a dynamic geological history resulting from the interplay of the North American, Cocos and Caribbean tectonic plates (Molnar and Sykes, 1969; Marshall, 2007). It displays wide ranges of elevation and soil types, high volcanic and seismic activities as well as repeated marine regressions and transgressions (Brezonik and Fox, 1974; Horn and Haberyan, 1993; Umaña et al.,1999; Haberyan et al., 2003; Obrist-Farner et al., 2021). In this region, at least 16 biogeographical provinces have been recognized, based on terrestrial plant and animal taxa (Morrone, 2014).

Numerous studies have attempted to elucidate the relationships between geodiversity and biodiversity and recognize drivers that promote current biological structure and biogeographical patterns in the northern Neotropics (Wallace, 1853; Patton et al., 1994; Gillespie and Roderick, 2014). Most evidence from terrestrial taxa suggests an *in situ* diversification, resulting from repeated events of colonization by North and South American taxa, prior to and after the closure of Panama isthmus estimated to have occurred between 15-4 Mya (Bacon et al., 2015; Montes et al., 2015). Molecular evidence suggests that extant

Mesoamerican terrestrial taxa (i.e. angiosperms, fern, birds, reptiles and mammals) originated primarily from the Amazonas Basin, with ancestors arriving by dispersion during the last 10 Mya (Antonelli et al., 2018).



These large-scale species movements across the American continent were mainly associated to Pleistocene large-amplitude climate fluctuations such as glacial and interglacial cyclicity and episodes of centennial to millennial persistence such as Late

Glacial, Heinrich and Dansgaard-Oeschger stadials (Behling et al., 2000; Carnaval and Moritz, 2008; Bouimetarhan et al., 2018; Baker et al., 2001, 2020). In terrestrial taxa, to both, geodiversity and climate change is then, mainly attributed current patterns of diversity, endemicity and distribution, on the basis of models such as biomes expansion and contraction (Smith et al., 2012; Leite et al., 2016), Pleistocene refugia (Haffer, 1969; Peterson and Nyári, 2008) and niche conservationism theory (Wiens and Donoghue, 2004).

In aquatic environments of the northern Neotropics, diversification and drivers for species distribution are by far less well known than those operating in terrestrial environments. For instance, most systems in this region are isolated and influenced either by volcanism or marine environments, which makes them highly variable (Albert and Reis, 2011; Bagley and Johnson 2014; Cabassi et al., 2019). Currently, the anthropogenic influence on aquatic environments and biological structure of the northern Neotropics has resulted in higher variability and deep changes on their pristine conditions (Albert and Reis 2011;

Wehrtmann et al., 2016; Franco-Gaviria et al., 2018). Most aquatic environments are used as water source for cities and agriculture; in large lakes, fishing and aquaculture production are causing eutrophication and invasive species introduction (i.e. *Oreochromis niloticus*). In addition, it is common that lakes are used as final dispose sites of waste waters, agrochemicals and mining residuals (McKaye et al., 1995; Soto et al., 2020).

Aquatic systems of the northern Neotropics are therefore, suitable to evaluate past and present influences of geodiversity,

climatic fluctuations, and anthropogenic influence for current configuration of regional species pool (Genner and Hawkins, 2016). Influence of geodiversity can be illustrated by the correlation between landscapes attributes with species ranges, based on the assumptions of species adaptations to local environments (Mahler et al., 2010; Arbour and López-Fernández, 2016). The influence of climate fluctuations and anthropogenic influence can also be tracked back based on the availability of fossil and subfossil species remains in sedimentary successions and then facilitate the understanding of historical, biogeographical

and ecological interactions such as migrations, colonization, speciation and extinction at local and regional scale (Correa-Metrio et al., 2012; Díaz et al., 2017; Cohuo et al., 2020; Pérez et al., 2021). The greatest limitation for using aquatic taxa to infer evolutionary traits in the northern Neotropics, is the scarcity of integrated and comparable regional studies displaying well-suited spatial and temporal limnological and biological data.

Freshwater ostracodes are a key group to evaluate such evolutionary traits in the northern Neotropics. Ostracodes are bivalved

microcrustaceans, that in recent environments are abundant, diverse and widely distributed (Pérez et al., 2011b, 2013a; Cohuo et al., 2016, 2020; Macario-González et al., 2018; Echeverria-Galindo et al., 2019). This species group shows levels of endemism as high as 74%, with distributions ranging from single lakes to the entire region (Cohuo et al., 2016). In sedimentary sequences of the northern Neotropics, ostracodes are abundant, particularly during the early Quaternary and late Pleistocene (Pérez et al., 2011b, 2013a; Cohuo et al., 2020).

In this study, we conducted an extensive survey of aquatic ecosystems across the northern Neotropics ranging from southeast Mexico to Nicaragua to provide large-scale limnological data and to answer two main scientific questions: (1) To what extent





is geodiversity controlling limnological properties and confining limnological regions?; (2) how does geodiversity (water and sediment geochemistry as integrals of geology and climate) affect the composition and distribution of ostracode species associations?


## 2. Material and Methods

### 2.1. Study area

Our study area covers the northernmost Neotropics, ranging from southern Mexico (Yucatán Peninsula) to Nicaragua (Fig. 1). Ecologically, this region is considered a biodiversity hotspot (Mesoamerican hotspot; Myers et al., 2000), with more than 5000

vascular endemic plants (De Albuquerque et al., 2015), about 1,120 bird species (~200 endemic), 440 mammal (65 endemic), 690 reptiles (~240 endemic), 550 amphibians (~350 endemic), and 500 fish species (~350 endemic) (Critical Ecosystem Partnership Fund). On this region also converge species representatives of both North and South America (Myers et al., 2000; Ojeda et al., 2003; DeClerck et al., 2010; Rull, 2011). The orography of the region is highly irregular with elevations ranging from sea level to more than 4500 m a.s.l. (Molnar and Sykes, 1969; Marshall et al., 2003, 2007). Plate tectonic interaction is

responsible for active volcanism (Central American Volcanic Arc) and high seismic activity (Marshall et al., 2003, 2007). The climate is typically tropical (Köppen-type group A-climate "tropical/megathermal climate", Peel et al., 2007) and predominantly warm (26°C mean annual temperature) (Waliser et al., 1999). Due to the irregular orography, at least a dozen climatic sub-zones are distinguished (Taylor and Alfaro, 2005). Precipitation depends primarily on the seasonal migration of the Intertropical Convergence Zone (ITCZ). The northern position of the ITCZ during summer results in the so called "rainy

season" in which precipitation can increase from 900 to 2500 mm yr$^{-1}$ (Hastenrath, 1967; Magaña et al., 1999). The hurricane season extends from July to December and is an important contributor to the humidity budget because on average 300 mm/day of atmospheric water can precipitate during tropical storms and hurricanes (Jury, 2011). The region is rich in water systems that can be of variable origin, shape, and hydrological dynamics, as well as in sediment, chemical and water composition. The karst Yucatán Peninsula, for example, with its 8000 cenotes, several lagoons, two major lakes (Chichankanab and Lake

Bacalar) and subterranean rivers is considered a unique hydrological region (Schmitter-Soto et al., 2002a, Alcocer and Bernal-Brooks, 2010). Whereas in Central America the lakes and wetlands cover more than 8% of the total land area (Ellison, 2004). Most important aquatic ecosystems in the northern Neotropics include coastal, tectonic and volcanic lakes (crater lakes and maars), karst waterbodies including lakes, cenotes and aguadas (water accumulated in topographic depressions under canopy cover); flooded caves, (subterranean) rivers, permanent and ephemeral ponds (Brezonik and Fox 1974; Pérez et al., 2011a;

Delgado-Martínez et al., 2018; Echeverría-Galindo et al. 2019; Obrist-Farner and Rice, 2019).





**Figure 1:** Simplified geological map of the northern Neotropical region showing the locations of the 76 studied aquatic ecosystems. Colors indicate geological units based on bedrock type and age of the sediments. Geological data was obtained from Garrity and Soller (2009). Black dots and numbers represent sampling localities. Detailed information of sampling sites can be found in Table 1. Legend: K_ Cretaceous sedimentary rocks; Kg_ Cretaceous Plutonic rocks; PZ, PZvf, PZx_ Paleozoic sedimentary rocks; Q_ Quaternary sedimentary rocks; Qvf, Qvm, TQv_ Quaternary volcanic rocks; T_ Tertiary undetermined age sedimentary rocks; TRJ_ Jurassic sedimentary rocks; eT_ Eocene sedimentary rocks; mT_ Miocene sedimentary rocks; mTvfi_ Miocene volcanic rocks; nT_ Eocene sedimentary rocks; oT_ Oligocene sedimentary rocks; paT, pgT_ Paleocene sedimentary rocks.

## 2.2. Sampling aquatic environments in the northern Neotropics

A total of seventy-six aquatic ecosystems located in five countries across the northern Neotropical region (Fig. 1) were sampled during July-October 2013, coinciding with the rainy season in the region. These systems are situated on the Yucatán Peninsula Mexico (n=28), Guatemala (n=26), El Salvador (n=14), Honduras (n=5) and Nicaragua (n=3) (Fig. 1). For all water bodies,





physical and chemical variables (temperature, dissolved oxygen, pH, conductivity) were measured *in situ* with a WTW Multi
Set 350i multiparameter probe at a water depth of 0.5 m. Maximum water depth of each site was measured with the
echosounder Fishfinder GPSMAP 178C. Location of sites, including elevation, latitude and longitude, was determined with
the navigator Garmin GPSmap 60c.

Water samples for analysis of major anions ($Cl^-$, $SO_4^{2-}$, $CO_3^{2-}$, $HCO_3^-$) and cations ($Ca^{2+}$, $K^+$, $Mg^{2+}$, $Na^+$) were collected at
water depths of 0.5 m below surface using a Ruttner-type sampling water bottle. All water samples were filtered in situ using
a 0.45µm pore size Whatmann glass microfiber filter. For cation analysis, filtered samples were acidified with $HNO^3$ to pH 2.
Waters were stored under refrigeration until analysis.

Biological samples were collected from the littoral zone and deepest bottom. At littoral areas, we sampled between submerged
vegetation using a 250 µm mesh hand net. Sediment samples were collected from the deepest part of the systems with an
Ekman grab, but only the uppermost centimeters of each grab were used for analysis.


### 2.3 Analysis of non-biological variables

Water chemistry analysis: Ionic composition was analyzed following Armienta et al. (2008). Bicarbonate was measured by
acid titration to pH 4.6, using a mixed indicator of methyl red and bromocresol green. Concentrations of calcium and
magnesium were obtained by complexometric titration with EDTA, whereas sodium and potassium were measured by atomic
emission spectroscopy. Chloride was potentiometrically determined using an ion selective electrode, adding a 5 M solution of
$NaNO^3$ as an ionic strength adjuster. Sulfates were determined by turbidimetry. Analytical quality was checked by ionic
balance, most samples balanced with less than 5% error. Major ions concentrations are expressed in mg/L, but these data were
also transformed to meq $L^{-1}$ and percentages to determine dominance of anions and cations and water type. Sodium and
potassium were summed. Ternary plots were constructed using the Past software (Hammer et al., 2001).

Sediment analysis: Total carbon (TC) and total nitrogen (TN) contents were determined by combustion under oxygen
saturation with a LECO TruSpec Macro CHN analyzer. Total inorganic carbon (TIC) was quantified with a Woesthoff
Carmhograph C-16 after dilution with phosphoric acid (45% H3PO4) and detection of the CO2-induced conductivity change
in NaOH. Total organic carbon (TOC) was calculated by subtracting TIC from TC. Qualitative and semi-quantitative
mineralogical compounds were examined by x-ray diffraction with a RIGAKU Miniflex600. For the identification and semi-
quantification of the minerogenic components the software Philips Highscore was used. All applied sediment analyses are
described in detail in Vogel et al. (2016).

Geological classification: The geological map of North and Central America generated by the Geological Society of America
(GSA) (Reed et al., 2005) and adapted and converted to a geographical information system (GIS) by Garrity and Soller (2009)
was used to identify geological regions in our study area. ArcGIS software was used to identify geological attributes of
sampling sites such as bedrock and age of sediments. Three major types of bedrock were distinguished: sedimentary rocks,
volcanites and plutonites. Ten geologic periods and epochs, respectively, were defined: Jurassic, Cretaceous, Tertiary



undetermined age, Paleogene undetermined age, Paleocene, Eocene, Oligocene, Neogene undetermined age, Miocene, and Quaternary.

**2.4 Limnological regionalization using cluster analysis and PCA**

We performed a cluster analysis (CA) to define groups of lakes based on the similarity of their measured attributes. For this analysis, we used a data set composed of 23 variables, out of which 21 were numerical, and the remaining two were categorical (Table S1). Numerical variables include altitude, limnological (water temperature, dissolved oxygen, pH, conductivity, $CO_3^{2-}$, $HCO_3^-$, $SO_4^{2-}$, $Cl^-$, $Na^+$, $K^+$, $Ca^{2+}$ and $Mg^{2+}$), sedimentological (total carbon (TC), total inorganic carbon (TIC), total organic

carbon (TOC), total nitrogen (TN)) and mineralogical (quartz, carbonate, phyllosilicates and feldspars) parameters. Categorical data are geological properties represented by bedrock type and age of sediments. We used the unweighted pair group method with arithmetic mean (PGMA) for CA, and Euclidean distance to investigate the grouping similarity of sampling points. Calculations were conducted in R software (R Development Core Team, 2015), using the vegan package (Oksanen et al., 2017).

We then used a Principal Component Analysis (PCA) for each of the main groups discriminated by the cluster, to identify correlated and explanatory variables of the data sets. For each group the first PCA run included 23 variables and those represented by superimposed arrows in the graphs were considered as evidence of correlation. Then, a second PCA run using uncorrelated variables was used to identify explanatory variables of the data sets. The PCA mix package implemented in R software (Chavent et al., 2014) was used because of its ability to handle quantitative and categorical data simultaneously. The

loading values for all parameters were obtained using normalized rotation.

To provide a graphical representation of most meaningful variables of the data sets detected in the PCA and further evaluate latitudinal and/or altitudinal environmental gradients, we created environmental variable-specific maps using kriging interpolation. We used an empirical semivariogram to quantify the spatial composition and structure of the feeding data (Wagner, 2003; Bivand et al., 2008). We then fitted a theoretical variogram using the nugget effect (0.5), and a partial sill (0.5)

in a linear model. We finally obtained a map representing measured data and estimates from unmeasured locations. The software Surfer® from Golden Software, LLC (www.goldensoftware.com) was used for calculations.

**2.5. Biological analysis: ostracodes**

Species identification and diversity metrics: Ostracode extraction and counting was carried out using 15 cm$^3$ of wet sediment.

Species identification was undertaken using three individual adult specimens with complete soft parts of each morphotype identified. Specimens were dissected using distilled water and glycerin (3%) under a stereomicroscope Leica MZ75. Selected shells from identified species were photographed with a Zeiss Axio Imager 2 microscope. Shells were stored in micropaleontological slides. Dissected soft parts were mounted on individual slides with Hydromatrix®. Soft part identification was conducted using species keys provided by Karanovic (2012). When keys did not fully resolve taxonomic

identities, original descriptions were consulted. Taxonomic classification follows Cohuo et al. (2016). Undissected material



was preserved in Eppendorf plastic vials with 70% ethanol and currently available at the ostracod collection from Instituto Tecnológico de Chetumal. Shannon diversity index was used for species diversity metrics, calculations were conducted using the Past software (Hammer et al., 2001).

Ostracode species associations: Species associations were examined by means of non-metric multidimensional scaling
(NMDS) (Legendre and Legendre, 1998). This procedure generates an ordination in a two-dimensional space, representing the pairwise dissimilarity between species according to their occurrences. We used the Bray-Curtis dissimilarity coefficient of presence-absence data (Sørensen Coefficient). Only species with at least two occurrences were included in this analysis. To test significance between species groups discriminated in NMDS, a permutational multivariate analysis of variance (PERMANOVA) was performed. We used a permutation with 9999 replicates and applied the Bonferroni correction.
Calculations were done with R software, using the vegan package (Oksanen et al., 2017).

Relating non-biological variables and ostracode abundances: The relative importance of geological settings on ostracode species composition was assessed by multivariate constrained ordination techniques. First, we performed a Detrended Correspondence Analysis (DCA) with detrending by segments and non-linear rescaling to estimate the extent of the environmental gradient (Hill and Gauch, 1980) and to decide which type of ordination would be more appropriate for our data
set. Species–environmental relationships were then analyzed using CCA (Ter Braak, 1986). Environmental variables were standardized and added by forward selection using the Monte Carlo permutation test with 999 permutations ($\alpha$ = 0.05). Calculations and final ordination graphs of the DCA and CCA were performed using the software Canoco version 5 (Šmilauer and Lepš, 2014).

## 2.6. Structural equation modelling

The influence of geodiversity and related environmental variables on species composition was assessed with structural equation modelling (SEM). This is a multivariate statistical technique that allows to model pre-defined causal relationships between observed and non-observed (latent) variables and test their statistical significance (Fan et al., 2016; Sarstedt and Ringle, 2020). Our conceptual model was based on the assumption that elevation gradients, latitude and bedrock, which are
all related to geodiversity, influence water chemistry and physical parameters of aquatic environments (limnology). Geodiversity and limnology are then assumed as direct or indirect drivers of freshwater ostracodes composition and diversity. Other factors, such as vulcanism, precipitation and marine-freshwater interactions that may determine species composition and diversity metrics in the northern Neotropics were also taken into account. We use major anion and cations as test variables.

Using a covariance matrix with a set of uncorrelated variables, we fitted various models using this conceptual framework. For
all models, statistical significance was tested with Root Mean Square Error (RMSE), comparative fit index (CFI) and standardized root mean squared residuals (srmr). The predictive power of the model (R-square) was measured based on the amount of variation of biological data. The most parsimonious model fitting our data set, was selected as explanatory model (Sect. S1 in the Supplement). The software R and the package Lavaan (Rosseel, 2012) was used for calculations.



# 3 Results

## 3.1 Limnological regionalization in the northern Neotropics

Cluster analysis identified two main groups representing limnological regions (Fig. 2). The first group (YG: Yucatán and Guatemala) consists of lowland lakes from the Yucatán Peninsula (Mexico), the Petén district (northern Guatemala) and the Pacific lowlands of southern Guatemala. The second group (GSHN: Guatemala, Salvador, Honduras, Nicaragua) consists of Guatemalan highland lakes, El Salvador and Honduras mid-elevation lakes and Nicaraguan lowland lakes.

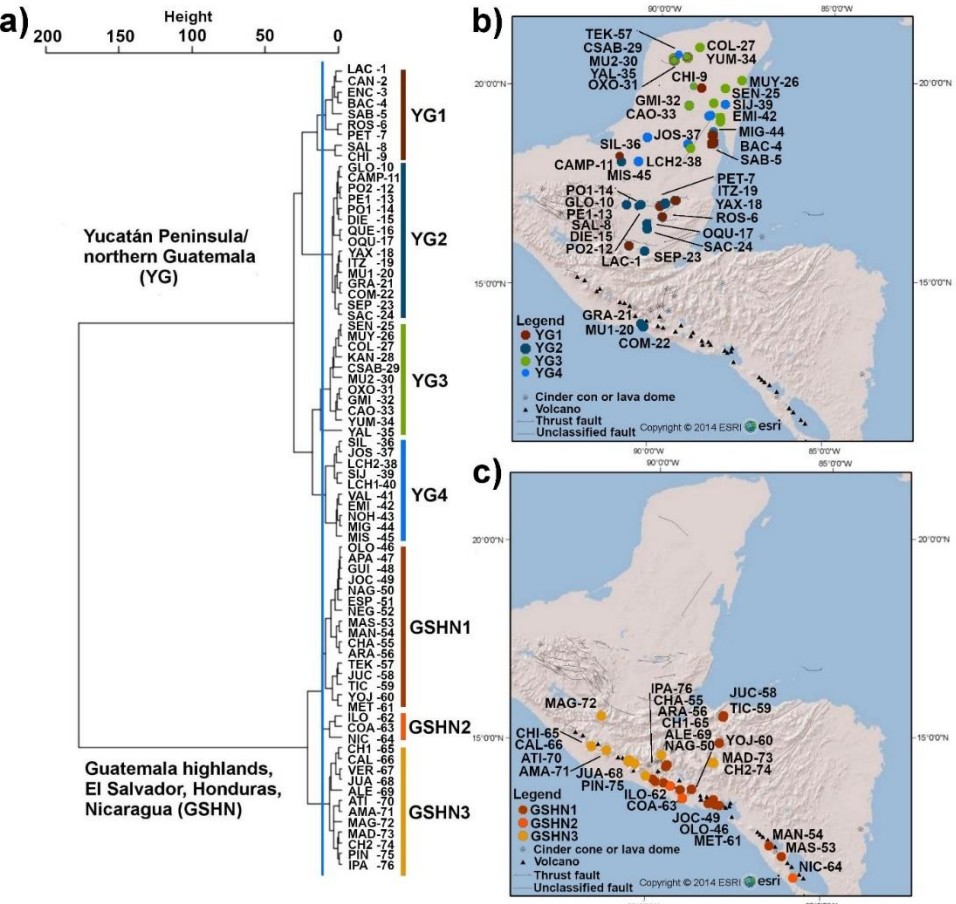

**Figure 2 (a) Cluster analysis dendrogram of the 76 studied aquatic ecosystems using 23 geographical, sedimentological, limnological and mineralogical variables. Blue line indicates cut-off criterion for cluster partition. Two major groups and seven subgroups were detected and named according to their position in the study area: Yucatán Peninsula/Northern Guatemala (YG) and Guatemala highlands, El Salvador, Honduras, Nicaragua (GSHN). In order to provide a graphic representation of cluster grouping, a color bar was assigned to each group and lakes within these groups were plotted on (b) YG map and (c) GSHN map using the same color. Lake full names of codes presented in cluster and maps are given in Table 1. Numbers of lake name codes correspond to that in Figure 1.**





Table 1 shows a list of all studied aquatic ecosystems located in YG and GSHN limnological regions and subregions, as well as information of selected attributes determined in this study. Detailed results of geological, limnological, sedimentological and mineralogical analyses for all studied water ecosystems can be founded in Table S1.

**Table 1. Limnological classification of 76 aquatic ecosystems of the northern Neotropical region, with their main geological, limnological, sedimentological and mineralogical properties.**

| Limnological group | Limnological subgroup | Country | Site number | Site name | ID | Coordinates | | | Elevation (m.a.s.l) | Surface area (km2) | Depth [a] (m) | Water type (ionic dominance) | Dominant minerals in sediments |
|---|---|---|---|---|---|---|---|---|---|---|---|---|---|
| | | | | | | N | W | Origin | | | | | |
| YG | YG1 | GUA | 1 | Lachua | LAC | 15.92 | 90.6732 | tectonic | 170 | 4 | 378 | SO4<<HCO3--Ca | Calcite |
| | | MEX | 2 | Candelaria river | CAN | 18.181 | 91.0493 | Tectono-karstic | 32 | river | 1.8 | HCO3<<SO4--Ca | Calcite |
| | | MEX | 3 | Encantada | ENC | 18.499 | 88.3895 | karst | 4 | 0.07 | 3 | SO4--Ca<<Mg | Calcite |
| | | MEX | 4 | Bacalar | BAC | 18.701 | 88.3819 | karst | 4 | 51 | 16 | SO4--Ca<<Mg | Calcite |
| | | MEX | 5 | Sabana de Chetumal | SAB | 18.517 | 88.3488 | karst | 7 | 0.88 | 1.6 | SO4--Ca--Mg | Nd |
| | | GUA | 6 | El Rosario | ROS | 16.526 | 90.1601 | karst | 126 | 0.02 | 3 | HCO3<<SO4--Ca<<Mg | Dolomite |
| | | GUA | 7 | Petexbatún | PET | 16.421 | 90.1862 | tectono-karstic | 120 | 5.6 | 40 | HCO3<<SO4--Ca<<Mg | Calcite |
| | | GUA | 8 | Salpetén | SAL | 16.981 | 89.6755 | tectono-karstic | 105 | 2.77 | 10 | SO4--Ca--Mg | Calcite |
| | | MEX | 9 | Chichancanab | CHI | 19.884 | 88.7682 | tectonic | 1 | 5.1 | 15 | SO4--Ca<<Mg | Calcite |
| | YG2 | GUA | 10 | La Gloria[b] | GLO | 16.955 | 90.3789 | tectono-karstic | 138 | 3.6 | 65 | SO4<<Cl--Ca | Nd |
| | | MEX | 11 | Camp | CAMP | 18.037 | 90.9887 | Flooded depression | 43 | 2.59 | 3.5 | SO4<<Cl--Ca | Nd |
| | | GUA | 12 | Las Pozas | PO2 | 16.343 | 90.1660 | karst | 152 | 2.16 | 35 | HCO3--Mg<<Ca | Nd |
| | | GUA | 13 | Petenchel | PE1 | 16.916 | 89.8288 | tectono-karstic | 118 | 0.73 | Nd | HCO3--Ca | Phyllosilicates |
| | | GUA | 14 | Poza azul | PO1 | 16.954 | 90.7884 | karst | 160 | 0.08 | 78.1 | HCO3--Mg<<Ca | Nd |
| | | GUA | 15 | San Diego[b] | DIE | 16.922 | 90.4231 | tectono-karstic | 156 | 3.8 | 8.1 | SO4<<Cl--Ca | Nd |
| | | GUA | 16 | Quexil | QUE | 16.923 | 89.8099 | tectono-karstic | 120 | 2.2 | 25.5 | HCO3--Ca | Nd |



|  |  | GUA | 17 | Oquevix | OQU | 16.651 | 89.7427 | karst | 150 | 1.6 | 10 | HCO3--Ca | Quartz |
|---|---|---|---|---|---|---|---|---|---|---|---|---|---|
|  |  | GUA | 18 | Yaxhá | YAX | 17.058 | 89.3897 | tectono-karstic | 164 | 7 | 22 | HCO3--Ca | Quartz and phyllosilicates |
|  |  | GUA | 19 | Peten itzá[b] | ITZ | 16.986 | 89.6938 | tectono-karstic | 112 | 100 | 165 | SO4<<HCO3--Ca<<Mg | Nd |
|  |  | GUA | 20 | El muchacho | MU1 | 13.889 | 90.1918 | Flooded depression | 3 | 0.36 | 2 | HCO3<<Cl--Na<<Ca | Feldspars |
|  |  | GUA | 21 | Grande | GRA | 13.890 | 90.1703 | Flooded depression | 5 | 0.95 | 2 | HCO3--Mg<<Ca | Phyllosilicates |
|  |  | GUA | 22 | Comandador | COM | 13.960 | 90.2544 | Flooded depression | 20 | 0.65 | 1.7 | HCO3--Ca--Mg | Phyllosilicates |
|  |  | GUA | 23 | Sepalau | SEP | 15.786 | 90.2167 | karst | 266 | 0.03 | 10.3 | HCO3--Ca | Quartz and phyllosilicates |
|  |  | GUA | 24 | Sacnab | SAC | 17.058 | 89.3725 | tectono-karstic | 170 | 4.28 | 9 | HCO3--Ca | Quartz and phyllosilicates |
|  | YG3 | MEX | 25 | Señor | SEN | 19.876 | 88.0775 | karst | 3 | 1.06 | 2 | Cl<<SO4--Na<<Mg | Calcite |
|  |  | MEX | 26 | Muyil | MUY | 20.075 | 87.6037 | karst | -1 | 2.52 | 16 | Cl<<HCO3--Na--Mg | Calcite |
|  |  | MEX | 27 | Colac | COL | 20.909 | 88.8669 | karst | 11 | 0.0002 | 120 | Cl<<HCO3--Na--Mg--Ca | Calcite |
|  |  | MEX | 28 | Kaná | KAN | 19.501 | 88.3954 | Flooded depression | 5 | 1.01 | 2.5 | HCO3<<SO4--Na--Ca | Nd |
|  |  | MEX | 29 | Sabak-há | CSAB | 20.580 | 89.5881 | karst | 18 | 0.0002 | 90 | Cl<<HCO3--Na<<Ca | Calcite |
|  |  | MEX | 30 | Mucuyche | MU2 | 20.624 | 89.6065 | karst | 17 | Cavern | 18 | Cl<<HCO3--Na--Ca | Ankerite |
|  |  | MEX | 31 | Oxolá | OXO | 20.678 | 89.2417 | karst | 18 | Cavern | 16 | HCO3<<Cl--Ca--Mg | Calcite |
|  |  | MEX | 32 | San Miguel | GMI | 19.935 | 88.9983 | karst | 32 | Cavern | 1.2 | HCO3<<Cl--Ca--Mg-Na | Calcite |
|  |  | MEX | 33 | Caobas | CAO | 18.445 | 89.1006 | Flooded depression | 126 | 0.13 | 4 | HCO3<<SO4--Ca | Calcite |
|  |  | MEX | 34 | Yumku | YUM | 20.578 | 89.6052 | karst | 16 | Cavern | 15 | HCO3<<Cl--Ca--Mg--Na | Calcite |
|  |  | MEX | 35 | Yalahau | YAL | 20.657 | 89.2170 | karst | 2 | 0.25 | 12 | Cl<<HCO3--Na--Mg | Nd |
|  | YG4 | MEX | 36 | Silvituc | SIL | 18.643 | 90.2727 | unknown | 69 | 7.9 | 3 | HCO3--Ca<<Na | Quartz and phyllosilicates |
|  |  | MEX | 37 | San Jose de la Montaña | JOS | 18.369 | 89.0120 | karst | 118 | 2 | 3 | HCO<<Cl--Na<<Ca | Nd |
|  |  | MEX | 38 | Chacanbacab | LCH2 | 18.478 | 89.0869 | Flooded depression | 109 | 257.29 | 3 | HCO3--Ca<<Na | Phyllosilicates |
|  |  | MEX | 39 | Sijil Noh ha | SIJ | 19.473 | 88.0554 | karst | 0 | 0.25 | 8 | SO4<<Cl--Na--Mg | Nd |





| | | Country | # | Name | Code | Lat | Lon | Type | | | | Hydrochemistry | Mineralogy |
|---|---|---|---|---|---|---|---|---|---|---|---|---|---|
| | | MEX | 40 | Chacchoben | LCH1 | 19.037 | 88.1811 | Flooded depression | 6 | 0.57 | 4 | HCO3<<Cl--Ca<<Na | Nd |
| | | MEX | 41 | Vallehermoso | VAL | 19.178 | 88.5216 | Flooded depression | 18 | 0.22 | 3 | HCO3--Ca<<Na | Quartz |
| | | MEX | 42 | Emiliano Zapata | EMI | 19.197 | 88.4691 | Flooded depression | 23 | 2.52 | 5 | HCO3--Ca<<Na | Quartz and phyllosilicates |
| | | MEX | 43 | Noh-bec | NOH | 19.146 | 88.1762 | Flooded depression | 1 | 8.5 | 2.5 | Cl<<SO4--Ca--Mg | Nd |
| | | MEX | 44 | Miguel Hidalgo | MIG | 18.786 | 88.3674 | Flooded depression | 31 | 20.22 | 4 | HCO3<<SO4--Ca<<Na | Calcite |
| | | MEX | 45 | Misteriosa | MIS | 18.042 | 90.4980 | unknown | 53 | 5 | 5.8 | SO4<<Cl--Ca | Quartz |
| GSHN | GSHN1 | SAL | 46 | Olomega | OLO | 13.307 | 88.055 | volcanic | 66 | 25.2 | 2.5 | HCO3--Na<<Mg | Phyllosilicates |
| | | SAL | 47 | Apastepeque | APA | 13.692 | 88.745 | volcanic, crater lake | 509 | 0.38 | 47 | HCO3--Mg<<Ca | Feldspars |
| | | SAL | 48 | Guija[b] | GUI | 14.261 | 89.501 | volcanic, lava flows | 431 | 45 | 22 | SO4<<Cl--Ca<<Na | Nd |
| | | SAL | 49 | Jocotal | JOC | 13.337 | 88.252 | volcanic, lava flows | 26 | 8.7 | 3 | HCO3--Na--Mg | Feldspars and phyllosilicates |
| | | SAL | 50 | Nagualapa | NAG | 13.470 | 89.002 | volcanic | 43 | 0.12 | 1 | HCO3--Na--Mg | Nd |
| | | SAL | 51 | El espino | ESP | 13.953 | 89.865 | volcanic | 689 | 0.99 | 5.5 | HCO3<<SO4--Mg<<Na | Phyllosilicates |
| | | SAL | 52 | Los negritos | NEG | 13.283 | 87.937 | volcanic | 102 | 0.69 | 2 | HCO3<<Cl--Na<<Mg | Phyllosilicates |
| | | NIC | 53 | Masaya | MAS | 11.996 | 86.116 | volcano-tectonic | 222 | 8.33 | ND | HCO3--Na<<Mg | Feldspars |
| | | NIC | 54 | Managua | MAN | 12.270 | 86.477 | volcano-tectonic | 41 | 1061.17 | ND | HCO3<<Cl--Na<<Mg | Feldspars |
| | | SAL | 55 | Chanmico | CHA | 13.779 | 89.354 | volcanic, crater lake | 477 | 0.78 | 51 | HCO3<<SO4--Mg<<Na | Feldspars |
| | | SAL | 56 | Aramuaca | ARA | 13.429 | 88.107 | volcanic, maar lake | 76 | 0.4 | 107 | SO4<<HCO3--Na<<Mg | Feldspars |
| | | MEX | 57 | Tekom | TEK | 20.73 | 89.4660 | karst | 24 | <0.01 | 3 | HCO3<<SO4--Ca<<Na | Calcite |
| | | HON | 58 | Jucutuma | JUC | 15.512 | 87.903 | Flooded depression | 27 | 4.34 | 2 | HCO3--Ca<<Na | Quartz and phyllosilicates |
| | | HON | 59 | Ticamaya | TIC | 15.551 | 87.890 | Flooded depression | 17 | 2.91 | 2 | HCO3<<Cl--Ca--Na | Nd |
| | | HON | 60 | Yojoa | YOJ | 14.861 | 87.985 | volcanic | 639 | 79.7 | 22 | HCO3-Ca | Quartz |
| | | SAL | 61 | Metapan | MET | 14.309 | 89.466 | volcanic | 450 | 16 | 6 | HCO3-Ca | Quartz |





| | | | | | | | | | | | | |
|---|---|---|---|---|---|---|---|---|---|---|---|---|
| GSHN2 | SAL | 62 | Ilopango | ILO | 13.682 | 89.071 | volcanic, caldera lake | 446 | 70.28 | 177 | Cl<<HCO3--Na | Nd |
| | SAL | 63 | Coatepeque | COA | 13.862 | 89.553 | volcanic, caldera lake | 743 | 26 | 119 | Cl<<HCO3--Na<<Mg | Feldspars |
| | NIC | 64 | Nicaragua | NIC | 11.460 | 85.774 | volcano-tectonic | 37 | 8264 | ND | HCO3<<Cl--Na<<Ca | Nd |
| GSHN3 | GUA | 65 | Chicabal | CH1 | 14.788 | 91.656 | volcanic, crater lake | 2726 | 0.21 | 10.3 | SO4--Mg | feldspars |
| | GUA | 66 | Calderas | CAL | 14.412 | 90.591 | volcanic, crater lake | 1790 | 0.35 | 26 | HCO3--Mg<<Ca | feldspars |
| | SAL | 67 | Verde | VER | 13.891 | 89.787 | volcanic, crater lake | 1609 | 0.1 | 12 | HCO3--Mg<<Ca | Feldspars |
| | GUA | 68 | San Juan Bautista | JUA | 14.042 | 90.072 | Flooded depression | 1285 | 0.07 | 2 | HCO3--Mg<<Ca | feldspars |
| | SAL | 69 | Alegria | ALE | 13.493 | 88.493 | volcanic, crater lake | 1272 | 0.09 | 10 | SO4--Ca<<Mg | Feldspars |
| | GUA | 70 | Atitlán | ATI | 14.684 | 91.224 | volcanic, caldera lake | 1556 | 125 | 340 | HCO3--Mg--Na | Nd |
| | GUA | 71 | Amatitlán | AMA | 14.436 | 90.548 | Volcanic | 1193 | 15.2 | 23 | HCO3--Cl--Na<<Mg | Calcite |
| | GUA | 72 | Magdalena | MAG | 15.543 | 91.396 | tectonic | 2863 | 0.01 | 2.8 | HCO3-Ca | Calcite |
| | HON | 73 | Madre vieja | MAD | 14.357 | 88.138 | Flooded depression | 1866 | 0.1 | 3.4 | HCO3--Mg<<Ca | phyllosilicates |
| | HON | 74 | Chiligatoro | CH2 | 14.376 | 88.183 | volcanic | 1925 | 0.04 | 5.5 | SO4--Mg | Quartz |
| | GUA | 75 | El pino | PIN | 14.345 | 90.394 | volcanic | 1038 | 0.64 | 6 | HCO3--Mg<<Ca | phyllosilicates |
| | GUA | 76 | Ipala | IPA | 14.557 | 89.639 | volcanic, crater lake | 1495 | 0.56 | 25 | HCO3--Mg<<Na | feldspars |

a= Maximum sampled water depth        b= Pérez et al., 2011        Nd=not determined


For YG region, the PCA with thirteen variables clearly explain the variation of the data set (Fig. 3a). The first (PC1) and second (PC2) components (dimensions) explain 38.1% of the total variance of the data set (Fig. 3a, Table S2.1). The PC1 accounts for 23.47%, and the PC2 for 14.63% of the total variance, respectively. The biplot based on dimension 1 and 2

indicates that conductivity and related ions Sodium ($Na^+$), Chloride ($Cl^-$) and Magnesium ($Mg^{2+}$) are the variables exhibiting the highest correlations (<0.64) with the first principal component. Thus, they represent the most influential variables differentiating aquatic ecosystems in the YG region. pH was highly correlated (>0.73) with the second component (PC2), suggesting that it is the second most influential variable of the YG aquatic environments (Fig. 3a, Table S2.1). Figures 3 b, c and d show regional distribution of the most meaningful variables for the YG region.





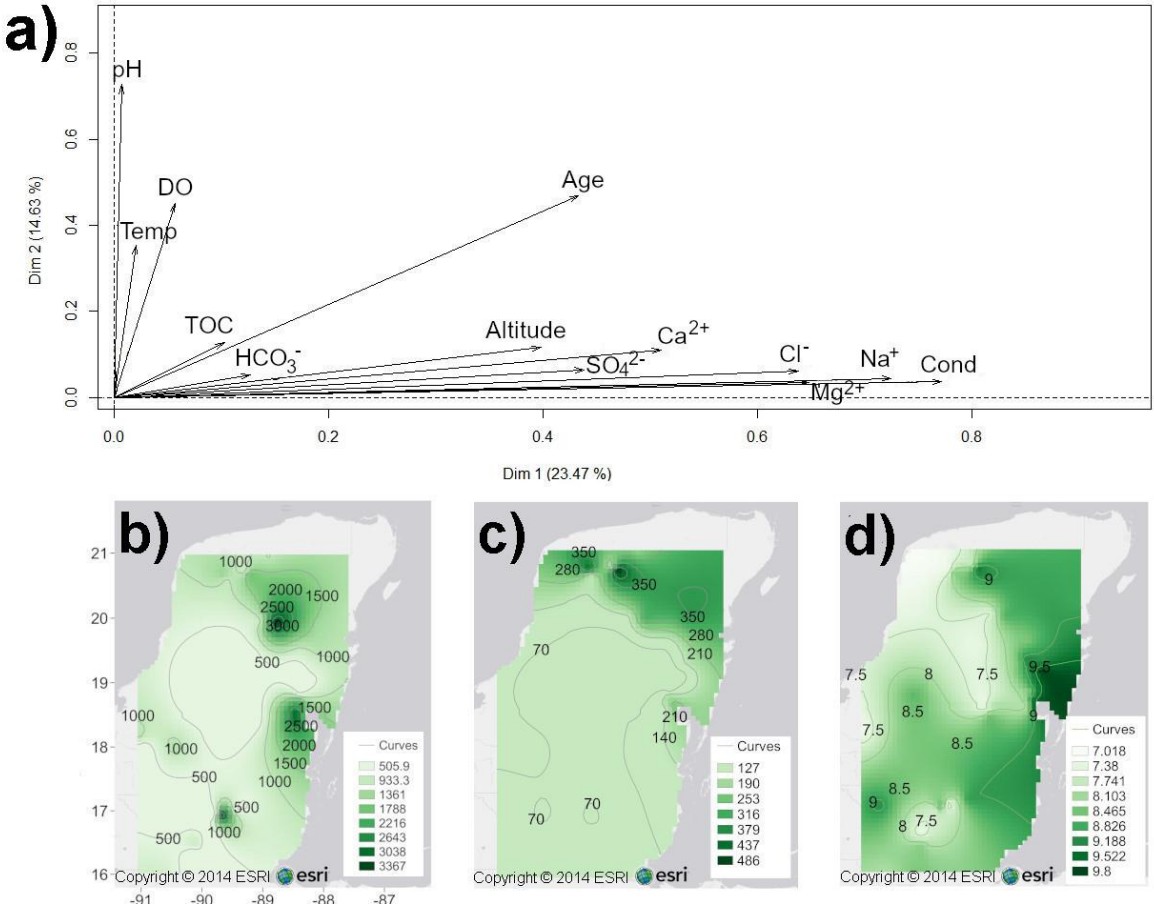

**Figure 3 (a) PCA biplot based on 12 variables from 45 aquatic ecosystems of the Yucatán Peninsula and northern Guatemala (YG limnological region). Conductivity and related ions were the most meaningful variables characterizing aquatic ecosystems as they explain 23.5% of the total variance. The second most important variable is pH, accounting for 14.63% of the total variance. Arrows represent variables. Spatial representations of meaningful variables are presented in (b-d) interpolated maps of conductivity (b); chloride (c); pH (d). Abbreviations are as follows: temperature (temp) (ºC), conductivity (cond) (μS/cm), magnesium (Mg$^{2+}$), chloride (Cl$^-$), bicarbonate (HCO$_3^-$), potassium (K$^+$), calcium (Ca$^{2+}$), sodium (Na+), sulfates (SO42-) dissolved oxygen (DO) (μmol/l) and total organic carbon (TOC). Major ions concentrations are expressed in mg/L.**

For the GSHN region, the PCA based on thirteen variables explain the 49.72% of the total variance of the data set, within the first (PC1) and second (PC2) components (Fig. 4a, Table S2.2). The PC1 accounts for the 26.83% and the PC2 for the 22.89% of the total variance. The PCA biplot based on dimension 1 and 2, respectively, shows that water ionic composition, particularly content of HCO$_3$, Na$^+$ and Cl$^-$ (correlated >0.62 with first component) is the most influential on discriminating aquatic systems in GSHN region (Fig. 4a, Table S2.2). Sedimentological and geological variables such as TOC, age and bedrock are the second most influential variables as they are strongly correlated with the second component (> 0.75) (Fig. 4a). Figure 4 b, c, d show the spatial distribution of meaningful variables for GSHN region.




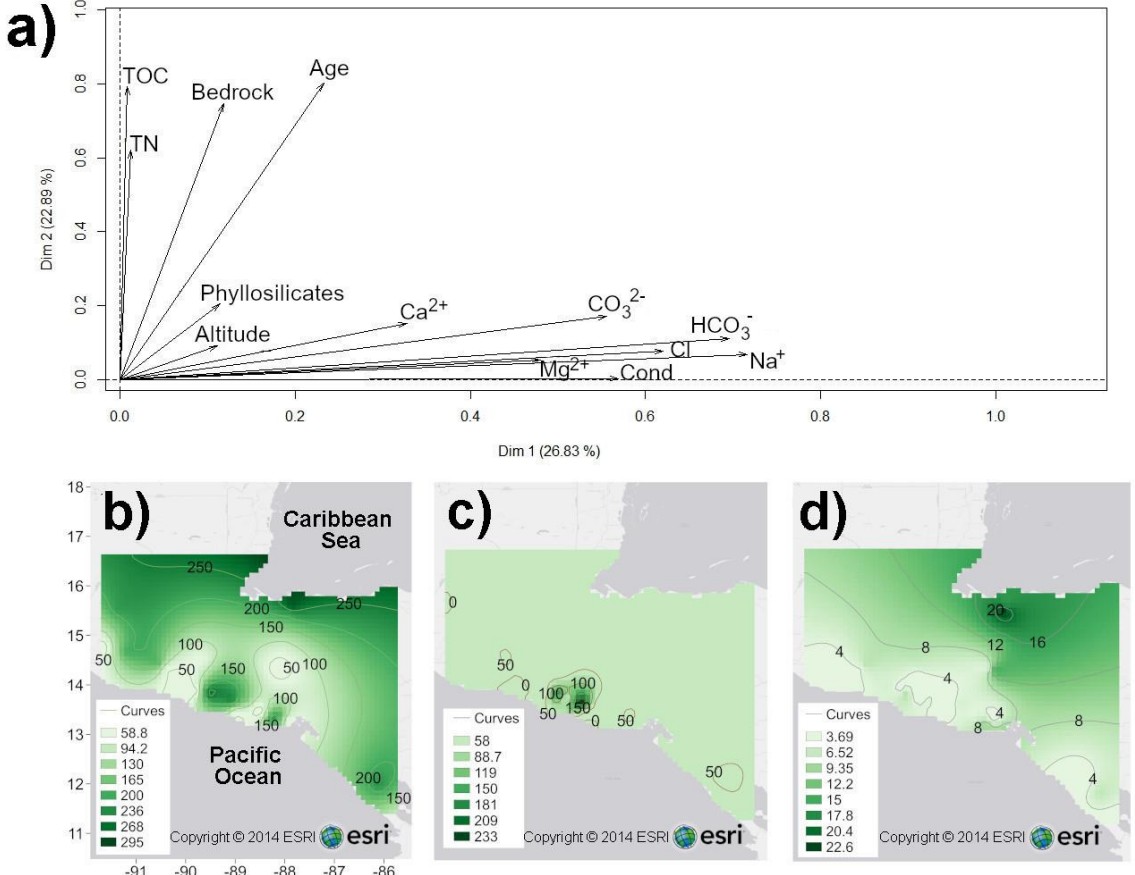

**Figure 4 (a) PCA biplot based on 13 variables from 31 aquatic ecosystems of central-southern Guatemala, El Salvador, Honduras and Nicaragua (GSHN limnological region). Ionic composition HCO$_3^-$, Na$^+$, Cl$^-$ were the most meaningful variables characterizing aquatic ecosystems as they explain 26.8% of the total variance. Sedimentology and geology were the second most important variables, accounting for 22.9% of the total variance. Arrows represent variables. Spatial representations of meaningful variables are presented in (b-d) interpolated maps of bicarbonates (b); sodium (c); TOC (d). Abbreviations are as follows: conductivity (cond) (µS/cm), magnesium (Mg$^{2+}$), sodium (Na$^+$), chloride (Cl$^-$), bicarbonate (HCO$_3^-$), carbonates (CO$_3^{2-}$), calcium (Ca$^{2+}$), total nitrogen (TN) and total organic carbon (TOC). Major ions concentrations are expressed in mg/L.**

Water ionic dominance was graphically evaluated with ternary plots (Fig. S1 and Fig. S2) and information on lake water types is shown in Table 1. Here, we highlight the most relevant characteristics for YG and GSHN limnological subregions:

### 3.2 The YG group composed by karst aquatic systems

The first limnological subgroup (YG1; n=9) within YG, included systems located in central-southern Yucatán and in the northern Petén district in Guatemala (Fig. 2). Lakes are located in lowlands (<170m a.s.l.) and most of them are relatively shallow (<16m depth) such as Bacalar, Encantada, Sabana Chetumal, Rosario, Salpetén, Chichancanab, except for Lake Petexbatún (40m depth) and Lachua (378 m depth). The latter constitute the deepest lake of the studied area. Waters of these



systems are dominated primarily by sulfates, followed by calcium and magnesium. Limnological subgroup YG2 (n=15) contains lakes that are mostly restricted to the Guatemalan lowlands (Fig. 2). Most of these systems are large lakes, including

Lake Petén Itzá, one of the largest (100 km$^2$), deepest (165m) and oldest (~400 ka) lakes of the northern Neotropics. Lake waters are dominated by carbonates, and therefore calcium prevails in these lakes. Aquatic systems located in the central and northern portion of the Yucatán Peninsula were grouped in YG3 (n=11) (Fig. 2). Colac, Sabak ha, Mucuyche, Oxolá, Gruta Miguel Hidalgo and Yumkú are cenotes from northern Yucatán, and lakes Yalahau, Caobas, Kaná, Señor and Muyil are located in the central-northern Peninsula. Carbonates dominate lake waters and chloride shows high values. Limnological subgroup

YG4 (n=10) consists of lakes located in the central-southern Yucatán Peninsula (Fig. 2). Lakes are relatively large, shallow and relatively far from the Caribbean and Gulf of Mexico coasts (>60km) such as Silvituc and Chacanbacab. Waters are dominated by carbonates and calcium.

Mineralogical analysis reveal that most lakes are dominated by carbonates in the YG limnological region. In subregions YG1 and YG3 (central and northern Yucatán Peninsula), most lakes sediments have calcite as dominant mineral (Table 1, Fig. S2)).

Lakes Chichancanab and Salpetén both YG1, however, show carbonates with a co-dominance of phyllosilicates and gypsum. Lakes from YG2 are mainly dominated by phyllosilicates and feldspars. The mineral composition of lake sediments of the YG4 subgroup varies. Sediments of the lakes such as Vallehermoso, Emiliano Zapata and Chacanbacab are dominated by phyllosilicates with or without feldspars. Lake sediments of lakes such as Miguel Hidalgo on the contrary, are dominated by carbonates with calcite as main mineral, whereas Lake Silvituc is characterized by exotic minerals such as silver and gold

(Table S1).

### 3.3 The GSHN group composed by volcanic aquatic systems

Lakes of GSHN1 limnological subgroup (n=16) are located in Central America at low and middle elevations ranging from 17 to 689 m a.s.l. Lake waters show a clear dominance of carbonates, followed by sodium, potassium and magnesium (Table 1, Fig. S3). Limnological subgroup GSHN2 is composed of three of the largest lakes in Central America: Ilopango, Coatepeque

and Nicaragua. These three lakes originated from volcanic activity. Lake Ilopango and Coatepeque are caldera lakes, whereas Lake Nicaragua is surrounding the volcanoes Concepción and Maderas. There is no clear pattern for ions in the water column, but sodium and potassium dominated, followed by magnesium. The limnological subgroup GSHN3 (n= 12) is formed mainly by crater lakes located in the highlands of Guatemala, Honduras and El Salvador. Most lakes waters are dominated by bicarbonates, followed by magnesium, while Chiligatoro, Chicabal and Alegría are dominated by sulfates (Table 1, Fig. S3).

Sediment mineralogy of Central American lakes (GSHN group) shows that most subgroups are dominated by feldspars. A co-dominance with other minerals such as phyllosilicates and carbonates occurs in the following subgroups. The GSHN1 combines lakes dominated by phyllosilicates and feldspars, whereas quartz is the dominant mineral in lakes Yojoa and Ticamaya (Honduras). GSHN2 include large lakes dominated by feldspars. In Lake Nicaragua, this dominance is also shared



with phyllosilicates. For the GSHN3, we detected two main mineral assemblages. The first is dominated by phyllosilicates and
feldspars, with clay minerals and feldspars as main minerals, and the second is dominated by feldspars and phyllosilicates.

### 3.4 Aquatic communities in the northern Neotropics

### 3.4.1 Ostracode species associations and its relationship with limnological regions

We found ostracode species in seventy-four out of the seventy-six aquatic systems of the northern Neotropics. In the Volcanic
Lake Alegría (El Salvador) and the karstic Cave San Miguel (Yucatán, Mexico), ostracodes were not observed. Living adult
specimens were encountered in samples from all systems, except for lakes Chicabal, Tekoh, Yaxhá, Verde and cenote
Mucuyche, where only empty shells or single valves were recovered. Taxonomic analysis of species allowed to identify seventy
species (Table S3), out of which thirty-one were recorded at single sites, whereas the remaining thirty-nine were observed in
at least two systems. Species richness ranges between 1 to 9 with an average of 4 species per site, whereas species diversity
Shannon index maximum value was H= 2.1, corresponding to Lakes Bacalar and Petén Itzá and River Candelaria. For all other
lakes this index average H= 1.1. A detailed list of ostracode species found in our study is presented in Table S3.
NMDS ordination based on species occurrence data, reveals five major species associations (OST 1-5) with a reliable stress
value of 0.08 (Fig. 5a) (Clarke, 1993). The PERMANOVA test further shows statistically significant differences between
groups centroids (F = 1.19, p = 0.0001), thus supporting NMDS ordinations. Ostracode group 1 and 2 are located in the YG
limnological region (karst terraces), and groups 3-5 in the limnological region GSHN (volcanic Guatemalan highlands and
Central American mid-elevations and lowlands). The first species group (OST1) consists of twelve ostracode species, recorded
from lakes and cenotes from the eastern Yucatán Peninsula (Fig. 5b). Most of these species are tolerant to high conductivities
(particularly related to Na+ and Cl-) such as *Heterocypris punctata* and *Limnocythere floridensis* (Fig. 5c) *Cyprideis* cf.
*salebros*a and *Perissocytheridea* cf. *cribrosa*. The second species group (OST 2) includes eleven species, distributed in lakes
and ponds of the southern Yucatán Peninsula and northern Guatemala (Fig. 5b). Some of these species (Fig. 5b), such as *Cypria*
*petenensis*, *Cypretta campechensis* and *Paracythereis opesta*, are considered endemic. Some others, such as *Alicenula*
*serricaudata*, *Pseudocandona antilliana* and *Cytheridella ilosvayi*, have a wide Neotropical distribution. The third species
group (OST 3) consists of seven species distributed mainly in mid-elevation lakes from Guatemala, El Salvador, and Honduras
(Fig. 5b). Several of these species have very restricted distributions and correspond to the *Strandesia*, *Keysercypria* and
*Cypridopsis* genera (Fig. 5c). The fourth species group (OST 4) is composed of six species: *Heterocypris nicaraguensis*,
*Potamocypris islagrandensis*, *Cypria granadae*, *Limnocytherina royi*, *Perissocythere marginata* and *Cyprideis* sp., distributed
exclusively in lakes in Nicaragua (Figs. 5a, b). The fifth group (OST 5) includes only three species from highland lakes of
Guatemala: *Chlamydotheca* cf. *colombiensis*, *Cypria* sp. 4 and *Cypridopsis* sp. 7 (Figs. 5b, c).


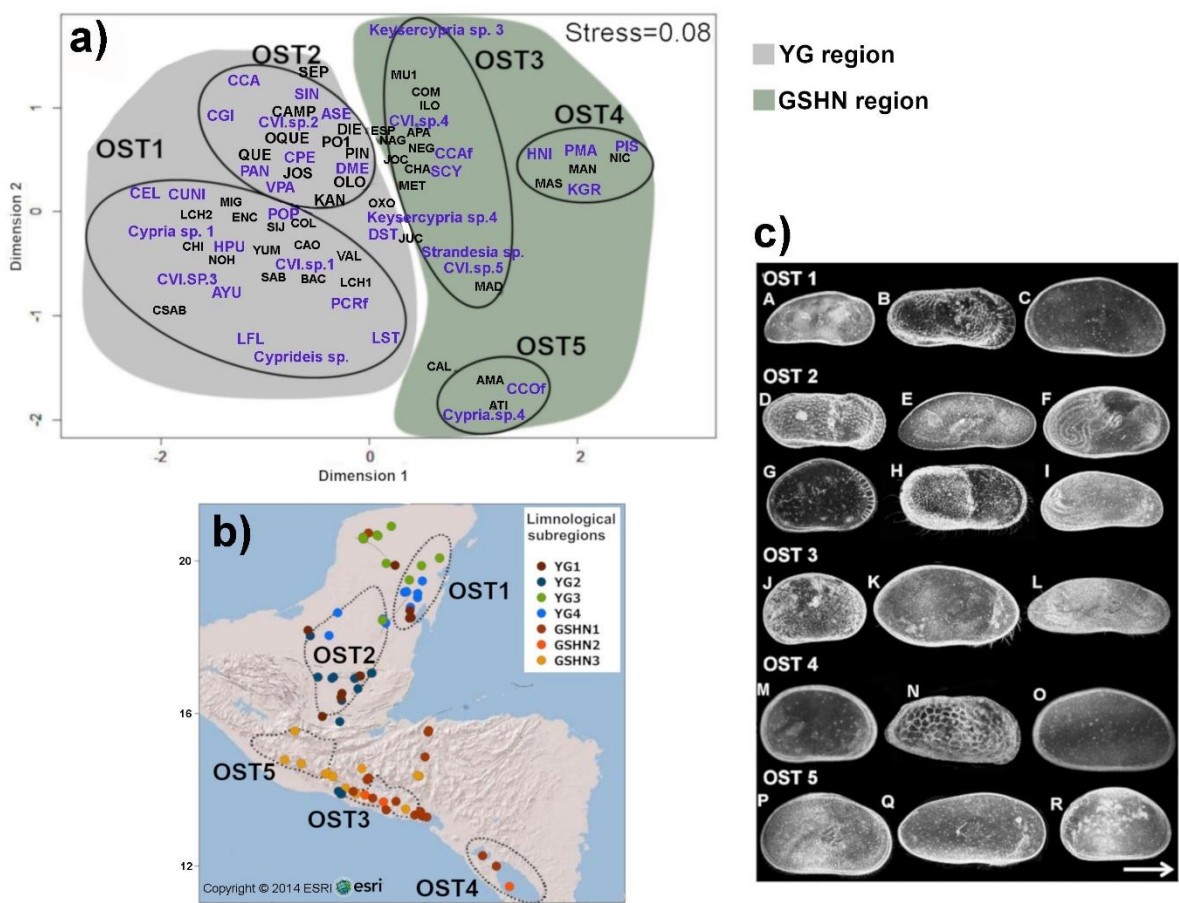

**Figure 5 (a) Non-metric multidimensional scaling (NMDS) plot showing differences in ostracode species composition among the 76**
**studied aquatic ecosystems. Results are based on Bray-Curtis dissimilarity index using species presence/absence data (stress = 0.08).**
**Black lines indicate ostracode group associations (OST 1, 2, 3, 4, 5). For species name abbreviations see Table S3. (b) map showing**
**spatial distribution of species associations. (c) Plate of selected freshwater ostracode species of each limnological subregion of the**
**northern Neotropical region. A)** *Thalassocypria* **sp.; B)** *Limnocythere floridensis***; C)** *Heterocypris punctata***; D)** *Paracythereis opesta***;**
**E)** *Diaphanocypris meridana***; F)** *Strandesia intrepida***; G)** *Cypretta maya***; H)** *Cytheridella ilosvayi***; I)** *Pseudocandona antilliana***; J)**
*Keysercypria* **sp. 4; K)** *Strandesia* **sp.; L)** *Stenocypris cylindrical major* **M)** *Cypria granadae***; N)** *Perissocytheridea* **cf.** *cribrosa***; O)**
*Heterocypris nicaraguensis***; P)** *Hemicypris* **sp.; Q)** *Pseudostrandesia* **sp.; R)** *Cypria* **sp. 4.**

### 3.4.2 Relating ostracode distribution with non-biological variables

Results of the DCA analysis reveal gradient lengths of 6.8 and 4.9 SD units for first and second axis, respectively. Thus,

a canonical correspondence analysis (CCA) was used to assess the relationship between species and geological and

environmental settings (Šmilauer and Lepš, 2014). Using the forward selection technique, we found a set of eight variables

that best explain the variance of data (Table S4.1). The first two axes of the CCA account for 57.8% of the variance of the

species-environment relationship and 16.2% of the variance of species data. The summary of performance and ordination of





this analysis are presented in Table S4.2. The first axis of the CCA plot (Fig. S3) explains 35.2% of the correlation between species and explanatory variables with a canonical coefficient of 0.93 and an eigenvalue of 0.68. Sedimentary bedrock,

temperature and feldspars are the most important factors that influenced species composition on axis 1 (Fig. S3). Axis 2 is strongly correlated with volcanic and plutonic bedrocks, conductivity, chloride and pH. This axis explains 22.6% of the species-environment relationship with a canonical coefficient of 0.83 and an eigenvalue of 0.44. In the CCA ordination, OST1 is positively correlated with conductivity and bedrock of sedimentary origin. OST 3 and OST 4 are positively correlate to volcanic and plutonic bedrock. OST 5 is associated with sediments dominated by feldspars. OST2 is positioned close to center

of the axes, suggesting that species are broadly tolerant for the variables analyzed.

### 3.4.3 Structural equation modelling

Six models of the influence of geodiversity, limnology and elevation on species composition were tested with structural equation modelling (SEM). These three variables were initially considered exogenous in the model and thus, explanatory for freshwater ostracode species distribution. Models using limnology and altitude as exogenous variables (in total 4 models),

were all rejected by p-values of Chi-squared test, much below of 0.05. When geodiversity and elevation were used as exogenous variable, SEM algorithm did not find a statistical solution for the models. Models that use geodiversity only as exogenous variable (two models) were retained as they display reasonable results. We provide details of all model performances in sect. S1 in the Supplement.

The optimal model (Fig. 6) is supported by the following metrics of global fit, CFI = 0.961 (values close to 1 indicate better

fit of the model), RMSEA = 0.01 (values less than 0.05 are considered reliable; Fabrigar et al., 1999) and SRMR=0.036 (value less than .08 is generally considered a good fit; Hu and Bentler, 1999). The optimal model suggests that geodiversity is the most influential for ostracode species distribution and composition through its indirect effect via limnology. Although the variable 'species associations' highly correlates with the variable 'species diversity', both were retained to explore the indirect influence of limnology on them, as they reflect different aspects of the biological structure of the region.

All paths in figure 6 are statistically significant (p-value <0.05), except for the direct effect of limnology on species associations. Conductivity account for the 67% of variance, whereas altitude explain 33% of the variance of the data set.





**Figure 6. Structural equation modelling of the influence of geodiversity and related variables with freshwater ostracode species composition and distribution. Metrics of global fit of the optimal model CFI = 0.961, RMSEA = 0.01, and SRMR=0.036. Arrows indicate the direction of influence. Black arrows are paths statistically significant at *p-value* <0.05, and gray path are not significant. Number next to arrows are the path standardized coefficients.**

## 4. Discussion

### 4.1 Geology and associated variables reveal two main limnological regions in the northern Neotropics

Our cluster analysis based on 76 lakes and 23 lake attributes shows a limnological regionalization of the northern Neotropics. Two main regions were identified, corresponding to Yucatán Peninsula-northern Guatemala (Group YG) and northern Central America (Group GSHN) (Fig. 2). The group YG is located in karstic terraces of sedimentary origin, dominated by limestone, dolomite, evaporites and carbonate-rich impact breccia (Hildebrand et al., 1995; Schmitter-Soto et al., 2002a, b; Vázquez-Domínguez and Arita, 2010). The Group GSHN is located in volcanic bedrock terrains of Guatemala, El Salvador, Honduras and Nicaragua, where pyroclastic and volcanic epiclastic materials, usually reworked, are abundant, reflecting active or past



volcanic activity (Dengo et al., 1970; Stoiber and Carr, 1973; Carr, 1984). The YG and the GSHN group, were further subdivided into four (YG1-4) and three subgroups (GSHN1-3) respectively, representing limnological subregions. This proposed regionalization therefore reveals high heterogeneity of aquatic systems in the northern Neotropics. Multivariate statistics (PCA) show that regions and subregions can be distinguished by ionic composition of waters. Sedimentological

parameters, such as TOC and mineral composition, are recognized as second most important characteristics (Figs. 3, 4).

In the YG karst region, lakes are characterized by carbonate-, calcium- and calcite-signatures, which is expected as waters interact with limestone and dolomite-rich sediments characteristics of the Peninsula bedrock (Schmitter-Soto et al., 2002a, b; Perry et al., 2009). This is also responsible for the water ionic dominance of sodium and magnesium, that in turn are related to a general pattern of alkaline surface waters in most aquatic systems of the region (Alcocer et al., 1998; Schmitter-Soto et al.,

2002a, b) (Figs. 3a, d). At specific areas (such as in the YG3), the dominance of chloride is also relevant, which can be explained by two main processes, 1) marine water intrusion and 2) subterranean waters having interacted with evaporites. The spatial distribution map of chloride contents in lake waters (Fig. 3c), shows a clear tendency to higher values in the northern Yucatán Peninsula and thus marine intrusion seems to be the most important source of chloride (Sánchez-Sánchez et al., 2015; Saint-Loup et al., 2018). For instance, sea water intrusions occur in northern Yucatán and were mapped to reach as far as 100

km inland (Steinich and Marín, 1996). Pérez-Ceballos et al. (2012) found that several water systems, mainly cenotes in this same region are characterized by marine waters below freshwater lenses, with water intermixing.

Sulfate is an interesting component of some lakes of the YG1 systems (Socki et al., 2002; Pérez-Ceballos et al., 2012). The presence of sulfates in lake waters may be attributed to the K/T anhydrite/gypsum–bearing impact breccia and dissolution of $CaSO_4$ of evaporites (Rosencrantz, 1990). This suggests that lakes with high sulfate contents receive high ground water input

as evaporites are only present at depths greater than 170 m below surface or that these lakes developed along sites with past tectonic activity such as faulting and uplift (Perry et al., 2002). Some lakes from the YG1 sub-groups with high content of sulfates did, in fact develop along fault zones. For example, lake Chichancanab is associated to the Sierrita de Ticul fault (Hodell et al., 2005) and Lachuá to the Polochic fault (Erdlac and Anderson, 1982), whereas others are lakes are related to high amount of ground water input, such as the Bacalar hydrological system (Perry et al., 2009).

High TOC values of lake sediments are important features of most lakes of sub-group YG2. These values are probably reflecting the trophic state of lake waters. Our data confirm results by Pérez et al. (2011a), who recorded TOC increase from north to south on Yucatán Peninsula. This may be attributed to a combined effect of soil, precipitation and vegetation type which change from north to south. The northern part of the peninsula is characterized by leptosols, which are shallow soils with high amounts of exposed hard rocks and calcareous material (Bautista et al., 2011; Estrada-Medina et al., 2013). There,

precipitation of about 450 mm a$^{-1}$ (Pérez et al., 2011a), allow the presence of small deciduous forests with overall low biological productivity. The south of the peninsula, where precipitation increase to values of >3,200 mm a$^{-1}$ (Pérez et al., 2011a), luvisols and vertisols, clayey and fertile soils support the growth of high evergreen forests providing high organic debris runoff, particularly during the rainy season.



Central American lakes are similar in origin as most of them are currently or were closely related to volcanism. These lakes are classified as caldera lakes, crater lakes in (partially) active or inactive volcanoes, maar lakes, or are located in volcanic bedrock basins (Golombek and Carr, 1978; Newhall and Dzurisin, 1988; Dull et al., 2001; Vallance and Calvert, 2003) (Table 1). The existence of at least three sub-regions highlights that these lakes are additionally influenced by regional factors related to orography (elevation), climate and the level of volcanic activity such as magmatic heat and gas input. Ionic dominance of

Central American lakes is highly variable, but sodium, potassium and magnesium are dominant, followed by chlorides, bicarbonates and sulfates (Fig. 4a). This ionic composition reveals two main processes controlling water chemistry, 1) active volcanic activity with strong interaction with lakes and 2) precipitation -evaporation rates, especially at high elevation lakes. Dominance of magnesium, chloride and sulfates, such as in GSHN3, can be attributed to active volcanic activity and hydrothermal systems. Chloride and sulfates are strongly influenced by volcanic gas input by incorporation of HCl and $SO_2$,

and interaction with igneous rocks. Lakes with such ionic dominance also display high contents of phyllosilicates and feldspars which may have formed by dissolution by hot and likely acid groundwater heated by hydrothermal activity. In central Mexico, lakes located in the Transmexican Volcanic Belt (TMVB) have (Armienta et al., 2008; Sigala, et al., 2017), similar ionic and sediment chemical composition than lakes in Central America. The ionic dominance of the crater lakes of volcano Popocatepetl (currently extinct by the intensification of volcanic activity in 1994), for example, changed from sulfate to calcium-magnesium

dominance and finally magnesium dominance, resulting from heating of andesite rocks after a period of increased volcanic activity (Armienta et al., 2000, 2008). This suggests that the influence of active volcanism on lake water chemistry may exert similar influences along the northern Neotropical region and American transition zone. High rates of evaporation are anticipated to represent an important driver for water chemistry in Central America as well, because of dominance of carbonates, bicarbonates and sodium (GSHN1 subregion). This ionic composition can be attributed to interaction and

weathering of volcanic rocks, capture and dissolution of $CO_2$ and high evaporation rates. Although Central America can be considered a tropical and humid region, high temperatures and solar radiation (especially at high elevations) may produce high evaporation rates leading to ion- specific signatures such as dominance of carbonates-bicarbonates.

Given the origin of discriminating variables in PCA for both YG and GSHN, we found three main sources controlling limnology of the northern Neotropics: (i) bedrock type, which determines specific mineral and ionic composition of lake

sediments and host waters; (ii) volcanic and marine influence, which determines the presence of dominant and conservative ions such as $Mg^{+2}$ and $Cl^-$ , and (iii) precipitation-evaporation balance across altitudinal and latitudinal gradients, which determines the concentration of solutes and therefore conductivity. This is consistent with results from limnological studies in other regions of Central America and southern Mexico, which suggested that geology (bedrock types and volcanic input to lakes) and marine interactions are the most influential for aquatic environments (Löffler,1972; Haberyan and Horn, 1999;

Haberyan et al., 2003; Cervantes-Martínez et al., 2002; Perry et al., 2002; Schmitter-Soto et al., 2002a; Socki et al., 2002; Pérez et al., 2011a). Several authors, however, consider additional features relevant for lake classification such as temperature, pH and altitudinal gradients (orography) (Brezonik and Fox, 1974; Horn and Haberyan, 1993; Umaña et al., 1999; Haberyan



et al., 2003). In our study, water temperature, pH and elevation received relatively low scores in the PCA, suggesting that their influence is relevant at local scale only.

Considering the determinants of limnological and sedimentological variability of aquatic systems at both regional and local scales, the interaction with marine environments and the high variability of aquatic systems morphology and origin, it becomes evident that current geological configuration is the main factor driving limnological properties and confining limnological regions in the northern Neotropical region.

**4.2 Geology as determinant of biodiversity and distribution in the northern Neotropics**

We found ostracode species in almost all systems and seventy species were recognized supporting the fact that this group is abundant and diverse in aquatic systems of the northern Neotropics. Interestingly, the number of species per lake is relatively low. In most sites we found between two to six species, and in large lakes of the region, such as Petén Itzá and Nicaragua, we found a maximum of nine species. Diversity metrics also recovered this tendency with values of Shannon index always below

of 2, which means low diversity in aquatic environments of the region (Margalef, 1957; Chao and Shen, 2003). Low species richness and diversity was also observed in previous studies in the region, for Lake Petén Itzá, for example, a maximum of eleven species was reported (Pérez et al., 2010a). In Lake Nicaragua, the largest in Central America, a maximum of seven species were found (Hartmann, 1959). Contrastingly, in regions such as the Palearctic, a single lake may host 32 species, such as in ancient Lake Ohrid (Macedonia/Albania) (Lorenschat et al., 2014), whereas small lakes in Europe, such as Lake

Maarsseen may harbor 9 to 15 species (Sluys, 1981). In Japan, forty species have been recorded from Lake Biwa (Karanovic, 2015). These global species richness patterns contrast with Latitudinal Diversity Gradient theory, that postulates increase of biodiversity from poles to the Ecuador (Mannion et al., 2014; Schumm et al., 2019). Interestingly, a similar pattern is observed in other aquatic microorganisms in the northern Neotropics, such as testate amoebae (Sigala et al., 2016; Sigala Regalado et al., 2018; Charqueño-Celis et al., 2019), chironomids (Hamerlík et al., 2018), and cladocerans (Vázquez-Molina et al., 2016;

Wojewódka et al., 2016). The low species diversity per lake in the northern Neotropics may be related to local environmental and evolutionary processes that have taken place recently in the geological time, namely during the Pleistocene and Holocene. For instance, in southern Mexico and northern Guatemala, severe droughts have been reconstructed based on sedimentary sequences of Lake Petén Itzá (Mueller et al., 2010; Cohuo et al., 2018). Some of these droughts, such those occurred at Heinrich Stadials are estimated that caused most shallow lakes desiccate at regional scale (Cohuo et al., 2018). Repetitive droughts in

the Pleistocene, with their consequent effect on surficial freshwater availability must have impacted in biological evolutionary processes of ostracodes (as benthic organisms), thus limiting species diversification and species adaptation. An alternatively explanation for the low ostracode species diversity in the northern Neotropics, rely in the human activity intensification in lakes. In Central America water extraction, fisheries and fish farms, eutrophication and wastewater disposal are among the most frequent alterations in freshwater ecosystem (Cuadra et al., 2006; Arcega-Cabrera et al., 2014; Campuzano et al., 2014).

These conditions are known to strongly affect aquatic taxa and particularly endemic species, thus causing loss of biodiversity (Moyle and Leidy, 1992). In the Neotropics, loss of biodiversity (genetic and phenotypic) in aquatic taxa is still under





development (Johann et al., 2019), but is certainly an important issue that must be considered for lake protection and management (Mercado-Salas et al., 2013).

Comparatively, in South America, in flood plains of the Upper Parana River, species richness can be as high as 44 species
(Higuti et al., 2017), interannually in this same system have been observed 38 species (Higuti et al., 2020), which is higher than that in lakes of Central America and comparable to richness of lakes in temperate regions. In Colombian aquatic systems, however, such as La Fé reservoir (Saldarriaga and Martínez, 2010) and River Magdalena basin (Roessler, 1990a, b) the number of species is again similar than that in Central America (~6 species per lake). Therefore, to clarify structural patterns of ostracods in the Neotropical region, more intensive sampling in lakes and rivers are needed. Evidence of other worldwide
tropical regions will be highly valuable to clearly understand patterns of ostracode diversity (tropical vs temperate) and how latitudinal diversity gradients act in the Ostracoda group.

The NMDS analysis shows the existence of at least five species associations in the region (OST1-OST5), emphasizing that ostracodes do not conform a faunal unit but rather disjunct faunas (Fig. 5a), similarly to that observed in freshwater fishes of the region (Miller, 1976; Matamoros et al., 2015). Ostracode associations are geographically delimited, and no overlap was
observed (Fig. 5b). Ostracode groups OST1 and OST2, belong to the YG limnological region, whereas OST3, OST4 and OST5 associations belong to GHSN limnological region (Figs. 5a, b). Few species were present in more than three limnological subregions, and these can be considered of wide Neotropical distribution, e.g. *C. ilosvayi* and *C. vidua*.

Correspondence between species associations and limnological subregions, suggest major influence of physical and chemical properties of lake environments over biological systems. The structural equation modelling (SEM) analysis in fact, recovered
the influence of limnology on species composition, but the direct effect of major and minor ions was not significant. When the limnology indirect effect over species composition is evaluated via conductivity, the SEM model provide significance and also resulted in 67% of the data variance observed. This implies that ostracodes species composition is more predictable using the overall conductivity, than using single ionic influence. SEM model also reveal that altitude is an important predictor of species composition (Fig. 6). They revealed a negative correlation, suggesting that elevation increase causes a decrease in the number
of species per site. Most highland lakes were characterized by up to three species and more commonly by a single species, except for the large lakes Amatitlán and Atitlán in Guatemala.

In all our SEM models, we were unable to find statistical significance for the direct effect of geodiversity on species composition (Fig. 6). This suggest that the influence of geodiversity may not be fully captured by our predictors, but the model using geodiversity as exogenous variable, was the unique that resulted in good metrics of fit. This reveals that the indirect
effect through limnology and elevation is significant.

The CCA, for example demonstrates that geological components of the landscape such as bedrock, elevation (temperature), minerology and ionic composition are important for species distribution (Fig. S3). For instance, the OST1 group is represented by lakes of the YG3 subregion associated to the Caribbean Sea coast (Fig. 5b). Similarly, the OST 2 corresponds to lakes of YG1 and YG2 subregions. These lakes are characterized by high TOC and abundant submerged vegetation. In Central America
patterns of coincidence between ostracodes and lake regions are evident as well (Fig. 5b), the OST 4 and OST 5 group are



associated to volcanic-influenced lakes in low (GSHN1) and highlands (GSHN3), respectively. Our SEM clearly shows the influence of geological features of landscape over limnological systems and species composition and distribution. This implies that changes in geodiversity or limnological attributes of the systems may cause drastic changes in composition of biological groups. Future biogeographic studies should focus on combining different zooplankton and zoobenthos groups.


## 5 Conclusions

The northern Neotropics is a region where relationships have historically been established between geodiversity, climate change and biodiversity. This study highlights that geodiversity has a linear and direct influence over limnological structures, and then indirectly for biological attributes of the aquatic systems of the region. Our limnological survey in 76 aquatic

environments identified two main limnological regions in the northern Neotropics. The YG region is associated to karst terraces in southern Mexico and northern Guatemala, whereas the GSHN is associated to landscapes formed by volcanic activity in southern Guatemala, El Salvador, Honduras and Nicaragua. At least seven limnological subregion were identified, illustrating high heterogeneity of aquatic systems of the region. Geodiversity and associated variables are primary drivers of lakes composition. Biological analysis of ostracode species assemblages shows that geochemical and limnological attributes of host

environments control species distribution. Low ostracode species richness and diversity (as evolutionary trait) in the northern Neotropics seem to be also strongly related to the geological history of the region. The low number of species per lake in the northern Neotropics contrasts with number of species per lake in temperate region which is at least five times higher. Questions about the applicability of the Latitudinal Diversity Gradient theory for Ostracoda and other zooplankton and zoobenthos groups remain open and sampling in the Neotropical and other tropical regions of the world is necessary. This is the first attempt to

integrate large scale limnology patterns in Central America. Further studies should focus on the establishment of a more detailed regionalization by including a higher number of lakes, environmental gradients and temporality.

*Data availability*. Geological, limnological, sedimentological and mineralogical data set of the 76 aquatic ecosystems sampled in this study are available in Table S1 in the Supplementary material and at the Pangaea repository: Data will be accessible pending manuscript acceptance. Ostracode relative abundances at each sampling sites can be found at the Pangaea repository:

Data will be accessible pending manuscript acceptance.

*Financial support*. Funding was provided by the Deutsche Forschungsgemeinschaft (DFG, SCHW 671/16-1), CONACYT (Mexico) through fellowships (218604, 218639) and Tecnológico Nacional de México, project number 9965.21-P.

*Author contributions.* LM and SC conducted the fieldwork, performed data analysis, and wrote the manuscript. AS, LP and ME managed and coordinated the project and contributed to manuscript writing. PH developed sediment analysis and gave

scientific input to the manuscript. MC developed water chemistry analyses and gave scientific input to the manuscript. AO, MP and MA provide support for sampling in their respective countries and processed sampling permits.



*Competing interests.* The authors declare that they have no conflict of interest.

*Acknowledgments.* We thank all colleagues and institutions involved in this work, including the student team: Christian Vera, León E. Ibarra, Miguel A. Valadéz and Cuauhtémoc Ruiz (Instituto Tecnológico de Chetumal, Mexico); Ramón Beltran (Centro Interdisciplinario de Ciencias Marinas, Mexico) and Lisa Heise (Universidad Autónoma de San Luis Potosí, Mexico) for their excellent contributions during field work. We would also like to thank the following colleagues and institutions that made samplings and water chemistry analysis possible: The team from the Asociación de Municipios del Lago de Yojoa y su área de influencia (AMUPROLAGO, Honduras); Eleonor de Tott, Roberto Moreno (Universidad del Valle de Guatemala, Guatemala); Consejo Nacional de Áreas Protegidas (CONAP, Guatemala); Néstor Herrera (Ministerio de Medio Ambiente, San Salvador), Teresa Álvarez (El Colegio de la Frontera sur, Chetumal Unit, Mexico); María Aurora Armienta (Laboratorio de Química Analítica, Instituto de Geofísica, Universidad Nacional Autónoma de México) and Adriana Zavala (El Colegio de la Frontera Sur, Mexico).

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
