# Peer review of "Geodiversity influences limnological characteristics and freshwater ostracode species distributions across broad spatial scales in the northern Neotropics"

_Biogeosciences, 2021_

## Referee Comment (RC2)

Review of Article

Title: Geodiversity primarily shapes large-scale limnology and aquatic species distribution in the northern Neotropics
Author(s): Laura Anahí Macario-González et al.
MS No.: bg-2021-298

**General comments**

This is an interesting study and this manuscript eventually will be worthy of publication. The authors are to be commended for their effort because this is a rather large data set from a poorly-studied part of the world.

Because the authors are evidently not native speakers of English, there are considerable numbers of grammatical mistakes and poorly-phrased passages, most of which are detailed below with suggestions for improvement.

In several places the text is over-generalized, for example Line 35 which refers to "aquatic biological composition" – in fact, this study is only about ostracods.

One area that requires clarification is the enumeratation of ostracods (Line 204). Apparently the investigators counted ostracods, but the details of counting aren't presented. How many per sample? It's also not clear what they mean by "identified three adults" passage. Analysis seems to be based on presence/absence, but if count data are available for each sample, then ordination can be based on square-root-transformed percentage data.

A more serious shortcoming is data handling. Apparently the investigators did not test each variable for normality. Skewed data should be transformed to produce a more normal distribution prior to ordination. For a clear methodology I recommend the most-recent edition of Tabachnick and Fidell's book *Using multivariate statistics*.

The authors should be careful in discussing diversity, which is not the same as richness. Also, such comparisons are difficult to interpret because collection size varies among the studies that they cite. Using 'alpha diversity' and 'beta diversity' would help.

I suspect that the authors are correct, that geology (local bedrock, karst vs volcanic) is the main influence on ostracod distribution. It's not clear why they use the term 'geodiversity' when bedrock geology alone seems to be the main driver. Data on 'geodiversity', i.e. local *diversity* of geology (geology, geomorphology, hydrology) aren't presented for each of the lakes.

Another driver seems to be understated, namely precipitation rates. Region YG is low rainfall, Region GSHN is generally higher rainfall. This correlates with karst/low elevation vs volcanic/highlands, but isn't rejected as a main driver. In order to conclude firmly that it's geology rather than rainfall, the authors should add another variable to their data set (i.e. mean annual rainfall at each lake).

**Specific comments**

- Line 1 the title is misleading – it indicates that shaping limnology is the primary result of geodiversity. Also, the article is about ostracods, not other aquatic species. Suggest "Geodiversity is a major factor in limnology and ostracod distribution in the northern Neotropics".
- Line 34 seems unsupported. I see no evidence of linear or continuous influence (if present, perhaps a figure could show this).
- Line 34 better words are available for the last sentence: "lake characteristics" not "limnological structure", "water chemistry" not "geochemical properties", and "ostracodes" not "aquatic biological composition".

- Line 120 not sure what is meant here. Maybe convert mm yr$^{-1}$ to mm month$^{-1}$, because the sentence is about the increased rainfall in the wet season, not in an entire year.
- Line 152 Were there multiple samples from the littoral zone of each lake? from the deepest part? Perhaps can indicate "Up to X samples were collected from littoral zones and up to Y samples from the deepest part".
- Line 153 can you give an approximate depth here, e.g. 30-60 cm ?
- Line 183 "elevation" not "altitude" throughout the article. "Elevation" refers to distance between sea level and the surface of the Earth; "altitude" need not be at ground level. So lakes have elevations, aircraft have altitudes.
- Line 204 I'm confused by the "counting". Later sentences indicate that only three adults were identified – did you identify three to determine species, then count until you had a larger number? Or did you only "count" the three that you identified? I do realize the difficulty in identifying species – would it be appropriate to identify them only to genus?
- Line 206 specify how ostracodes were isolated from the sediment. Direct 'picking' under a dissecting microscope?
- Line 250 Figure 1. In YG I see 39 localities but there are 45 lakes listed for YG. CAN-2, ENC-3, QUE-16, and KAN-28 seem to be missing from the map, and apparently two others. Also TEK-57 is shown on the YG map but is listed among GSHN in the cluster chart.
- Line 465 the cation list is first, the anion list is second. The phrasing here suggests that these anions are less common than these cations, but that indicates a charge imbalance. Ions should be listed by importance in a single list (e.g. Ca, Cl, Mg, Na… etc).
- Line 474 is the volcano "*dormant*" since 1994, or are its crater lakes "extinct" (whatever that means) as is suggested by the phrasing)? Please clarify. The phrase "extinct by the intensification of volcanic activity" makes no sense, because "extinction" cannot be caused by intensification of volcanic activity.
- Line 512 and elsewhere: use "richness" and "diversity" carefully, because they are not identical. "Richness" is the number of species, but "diversity" also factors in relative abundance. In addition, be aware of the difference between alpha diversity (e.g. in one lake) and beta diversity (e.g. in lakes of one region). In Line 515 you're clearly writing about alpha diversity, but Line 511 seems to be about beta diversity. The two are not equivalent.
- Line 505 and later: these data reflect, in part, collector effort (for example, Hartmann probably collected many samples from around L. Nicaragua). Thus it is hard to reach definite conclusions when comparing diversities.
- Line 530 it's not clear what "interannually… 38 species" means. Because 38 is less than 44, there seems to be no new information in this phrase.

**Technical corrections**
- Line 20 "ecosystem" not "ecosystems"
- Line 23 insert comma after "limnology"; throughout the paper, in a list of three or more words, insert a comma before "and" (e.g. "geodiversity, limnology, and aquatic species associations")
- Line 26 "karst" not "Karst"
- Line 27 change "mid-elevations" to "mid-elevation" and place between "volcanic" and "terrains"
- Line 28 insert "to" after "attesting"
- Line 28 change "identifies" to "identified"
- Line 28 change "ionic" to "chemistry"
- Line 29 remove comma after "formation"
- Line 31 add "s" to "association"
- Line 31 change "are" to "were"
- Line 32 change "reveals" to "revealed"

- Line 34 insert comma after "continuous" (as was done at Line 23)
- Line 35 insert comma after "properties" (as was done at Line 23)
- Line 35 change "Lakes" to "lakes"
- Line 40 insert "its" between "At" and "interaction"
- Line 40 change sentence from "contribute…. ecosystems" to "influence nutrient delivery to ecosystems"
- Lnie 45 change "is" to "results from"
- Line 45 insert comma after "composition"
- Line 48 change "Tropics" to "tropics"
- Line 54 change "central-southern" to "south-central"
- Line 56 delete comma after "types" and insert "along with" between "types" and "high"
- Line 63 replace "Panama" with "the Panamanian"
- Line 65 replace "Amazonas" with "Amazon"
- Line 66 delete "by dispersion"
- Line 68 replace "to" with "with"
- Line 71 change "In terrestrial taxa…. patterns" to "Therefore, for terrestrial biota, geodiversity and climate change have been the main influences on distributions"
- Line 72 change "on the basis of" to "following models of"
- Line 72 change "biomes" to "biome"
- Line 79 change "on" to "to"
- Line 80 change "source" to "sources"
- Line 82 change "dispose" to "disposal"
- Line 84 delete comma after "therefore"
- line 84 insert "places" between "suitable" and "to"
- Line 86 change "landscapes" to "landscape"
- Line 88 change "tracked back based on…facilitate" to "seen in fossil and subfossil communities, which improves"
- Line 93 change "well-suited" to "appropriate"
- Line 94 again, maybe some other word than "traits"
- Line 95 delete comma after "microcrustaceans", and move "in recent environments" to follow "distributed"
- Line 97 change "with distributions ranging from" to "with individual species distributed in"
- Line 101 change "To" to "to"
- Line 102 change "(water and sediment… climate)" to "(geology and climate, as reflected by water and sediment chemistry)"
- Line 110 change "vascular endemic" to "endemic vascular"
- Line 112 need a year after "Partnership Fund"
- Line 112 change "On this region… America" to "In addition, species have arrived from South America and from more distant North America"
- Line 120 put negative sign in the superscript: "mm yr$^{-1}$" not "mm yr-$^1$"
- Line 121 be consistent – use superscripts OR slashes throughout (e.g. mm day$^{-1}$ OR mm/day)
- Line 126 delete "Whereas" and change "in" to "In"
- Line 127 change "include" to "can be categorized as"
- Line 129 change "(subterranean) rivers" to "rivers (both surface and subterranean)"
- Line 129 insert "as well as" between "rivers," and "permanent"
- Line 133 change "data was" to "data were"
- Line 140 change title to "Field sampling"
- Line 142 insert "of" after "Peninsula"
- Line 145 change "the" to "a"
- Line 146 delete "echosounder"
- Line 147 change "the" to "a" and delete "navigator"
- Line 148 change "Water samples… Na$^+$)" to "Sample for water chemistry".

- Line 150 move superscript "3" to be a subscript
- Line 152 change "At" to "In"
- Line 157 recommended to avoid sub-sections here e.g. "Water chemistry analysis:" Instead, start the first sentence of each paragraph with a phrase of its data, e.g. for the first paragraph just delete "Water chemistry analysis". Often it makes sense to start this sentence with "For" (e.g;. "For sediment analysis, ") or "In order to".
- Line 161 make "3" a subscript
- Line 162 change "balance," to "balance;"
- Line 164 change "Past" to "PAST" and give version number
- Line 180 change title to "Cluster analysis and PCA of lake groups"
- Line 187 capitalize "Vegan"
- Line 190 change "for each… cluster" to "for each cluster" and delete comma after "cluster,"
- Line 191 insert comma after "variables"
- Line 192 change "arrows" to "vectors"
- Line 196 delete "of the data sets … altitudinal" to "and to evaluate environmental"
- Line 198 by "feeding data" do you mean "input data" ?
- Line 201 transpose "software" and "Surfer®"
- Line 203 suggest title be "Ostracode analysis"
- Line 211 insert "are" between "and" and "currently"
- Line 212 change "used for …. using the Past" to "calculated using PAST"
- Line 220 change "vegan" to "Vegan"
- Line 227 delete "the software"
- Line 232 change "to model" to "the modelling of"
- Line 234 change ", which are… geodiversity," to "(which are related to geodiversity)"
- Line 236 change "assumed as" to "taken as"
- Line 237 start the sentence with "We aslo considered other factors," and delete "were also taken into account"
- Line 238 "used" not "use"
- Line 241 "R-squared" not "R-square"
- Line 242 delete comma after "data set"
- Line 245 change title to "Lake regions"
- Line 257 change "Lake full names of codes … Figure 1." to "Lake names are given in Table 1 and correspond to those in Figure 1."
- Line 262 change "founded" to "found"
- Line 265 Table 1 could be presented more clearly. Align the column of Site names to the right, not centered. Reduce width of the first column. Use hyphens in column titles where appropriate. Increase width of coordinates so that digits fit on a single line. Make "Origin" a full column on its own (it is not, after all, a coordinate). Use abbreviations where helpful (e.g. T for tectonic, Mx for Mexico) and define these (and others like "Nd") in the table's caption. Transpose "Country" and "Site number" columns. It may be necessary to reduce the font by 1 or 2 units to prevent wrapping of cell contents. Please check the use of "<<" as it seems reversed: for example, as presented Site 2 has much less bicarbonate than sulfate.
- Line 265 Table 1 it may be worthwhile to separate these data into two tables, one for physical features (country, lat/long, elevation, surface area) and one for other features (with quantitative data for ions, e.g. Mg and Ca in their own columns). It may help to list the common ostracods in each lake here as well.
- Line 271 use past tense throughout, e.g. "explained" not "explain" and "were" not "are"
- Line 275 use lower-case for element names (e.g. "magnesium" not "Magnesium") throughout the paper
- Line 283 be consistent with two decimal places in these percentages
- Line 308 suggest title be changed to "The YG lakes (karst terrain)"
- Line 309 rephrase "subgroup (YG1; n=9… central-southern," as "subgroup in Region YG was

Subgroup YG1 which included nine lakes in south-central"
- Line 315 please provide reference to the age of the lake
- Line 315 again use past tense in most places in this paper.
- Line 319 "north-central" not "central-northern" and make similar changes throughout the paper
- Line 324 change "))" to ")"
- Line 325 put "both YG1" in parentheses
- Line 331 suggest title be changed to "The GSHN lakes (volcanic terrain)"
- Line 346 delete "in the northern Neotropics"
- Line 347 change "its" to "their"
- Line 351 change "to identify" to "the identification of"
- Line 353 change "whereas… value was" to "whereas the maximum value of the Shannon diversity index was"
- Line 355 insert "was" after "average"
- Line 356 delete "baseed on species occurrence data,"
- Line 359 delete "Guatemalan… American"
- Line 382 suggest "Relating ostracodes to environmental variables"
- Line 390 "Axis 1" not "axis 1"
- Line 398 change "thus," to "thus were"
- Line 399 change "altitude" to "elevation" here and elsewhere
- Line 407 delete "and composition"
- Line 410 "Figure" not "figure"
- Line 411 "accounts" not "account"
- Line 415 "p" not "*p-value*"
- Line 415 "and the gray path is", not "and gray path are"
- Line 416 "Numbers" not "Number"
- Line 426 change "GSHN group," to "GSHN groups"
- Line 432 "peninsula" not "Peninsula"
- Line 433 delete "water ionic" and insert "in the water" after magnesium
- Line 435 delete "in the"
- Line 436 replace "and" with "and/or"
- Line 437 delete "spatial" and "map" and "contents in lake waters" and the comma after "3c)"
- Line 438 replace ""seems to be the most" with "may be the more"
- Line 441 insert comma after "region"
- Line 449 "amounts" not "amount"
- Line 450 "subgroup" not "sub-group" – or change all to "sub-group" in the paper
- Line 455  units of "a$^{-1}$" are given as "yr$^{-1}$" elsewhere.  Please be consistent.
- Line 455 delete comma after "2011a)" and change "allow" to "allows"
- Line 456 "increases" not "increase"
- Line 460 insert "Many" before "Central American"
- Line 465 insert "often" after "are"
- Line 466 replace comma with colon
- Line 467 delete space in "precipitation -evaporation"
- Line 472 delete comma after "2017)" move "(Armienta… 2017)" to follow "Central America"
- Line 473 change "than" to "as"
- Line 485 insert "to" after "finally"
- Line 478 replace "because of" with "as evidenced by the"
- Line 482 delete space in "ion- specific", and replace "carbonates-bicarbonates" with "carbonates and bicarbonates"
- Line 484 use "1)", "2)", and "3)" as you do elsewhere, rather than "i)", "ii)", and "iii)".
- Line 495 please rephrase this sentence, for example as "At regional and local scales, aquatic systems vary in limnology and sedimentology, and these are strongly influenced by morphology and origin.

Because morphology and origin reflect geological factors it is clear that, overall, geology is the primary determinant of lake properties and therefore also of geographic lake groups."

- Line 500 suggest title "Geology as driver of ostracode distribution"
- Line 501 change "ostracode species" to "ostracodes"
- Line 501 change "systems … Neotropics" to "systems, and the fact that we found seventy species confirms high richness for this group the northern Neotropics".  Is 70 species really considered a large number of species??
- Line 505 delete "of"
- Line 505 insert "individual" between "in" and "aquatic"
- Line 509 change "as in ancient" to "as ancient"
- Line 512 "equator" not "Ecuador"
- Line 513 change "occurred at" to "during"
- Line 514 replace "Stadials … scale" with "Stadials, caused most shallow lakes in this region to dry out completely"
- Line 515 suggest to start a new paragraph with "The low species diversity…."
- Line 520 insert comma after "availability"
- Line 520 replace "in biological evolutionary processes of" with "the evolution of"
- Line 521 delete "(as benthic organisms)" and delete "and species adaptation"
- Line 521 "alternative" not "alternatively"
- Line 522 delete comma after "Neotropics" and replace "rely in" with "relies on"
- Line 522 rephrase "human activity intensification in" as "intensified human impacts on lakes"
- Line 523 change "America water … ecosystem"  to "America, the greatest human impacts are frequently water extraction, fishing, fish farming, eutrophication, and waste water disposal"
- Line 525 delete "are known to"
- Line 526 insert "both" between "(" and "genetic"
- Line 527 change "development" to "investigation"
- Line 529 change "Comparatively…  44 species" to "In South America, the floodplains of the Upper Parana River may support up to 44 species"
- Line 531 change "In Colombian… however," to "However in Colombia"
- Line 534 change "are" to "is"
- Line 534 change "of other" to "from other" and delete worldwide
- Line 536 change "how …. group" to "and latitudinal diversity gradients"
- Line 538 change "do not… faunas" to "form distinct assemblages"
- Line 538 delete "ly" from "similarly"
- Line 540 rephrase sentence as "Ostracode assemblages in the YG region (OST1 and OST2) are distinct from those of the GHSN region (OST3, OST4, and OST5)."
- Line 543 delete comma after "subregions"
- Line 544 change "over" to "on"
- Line 544 replace "in fact, recovered" to "confirmed"
- Line 544 rephrase sentence as "The indirect effect of conductivity on species composition, however, was significant in SEM and explained 67% of the variation in the data.
- Line 548 delete comma after "conductivity"
- Line 548 inset "The" before "SEM" and change "reveal" to "reveals"
- Line 548 change "altitude" to "elevation" here and elsewhere in the paper.
- Line 549 rephrase as "The SEM models also suggest a reduction of species richness with elevation of individual sites." (this is alpha diversity)
- Line 553 change "suggest" to "suggests"
- Line 554 delete comma after "variable"
- Line 556 delete ", for example"
- Line 558 change "to" to "with"
- Line 559 change "In Central America .. evident" to "In the GHSN region ostracodes were also distinct

between subregions"
- Line 560 change comma after "5b)" a colon
- Line 561 change "to" to "with"
- Line 564 delete "Future …. groups."
- Line 567 change "structures" to "characteristics"
- Line 569 change"for" to "over"
- Line 573 change "of the" to "within each"
- Line 573 change "lakes composition" to "limnology"
- Line 575 change "trait" to "traits"
- Line 580 rephrase last sentence perhaps as "Further studies should aim to expand our knowledge of lake and biotic regions by including a greater number of lakes, variables, and taxonomic groups."

-

---

## Author Comment (AC2)

Biogeosciences Discuss., author comment AC1
https://doi.org/10.5194/bg-2021-298-AC1, 2022 ©
Author(s) 2022. This work is distributed under the
Creative Commons Attribution 4.0 License.

[Figure]

**Reply on RC2**

Laura Anahí Macario-González et al.

Author comment on "Geodiversity primarily shapes large-scale limnology and aquatic species distribution in the northern Neotropics" by Laura Anahí Macario-González et al., Biogeosciences Discuss., https://doi.org/10.5194/bg-2021-298-AC1, 2022

**Reviewer 2.-** We appreciate all the comments and suggestions, they will significantly improve the manuscript. Below we provide answers to all comments. We have tried to address most of the suggestions made by reviewer 2, but in case this is not possible, we provide a detailed explanation.

Comment: Because the authors are evidently not native speakers of English, there are considerable numbers of grammatical mistakes and poorly-phrased passages, most of which are detailed below with suggestions for improvement.

Answer: Sentence structure and grammatical errors will be corrected by a professional English editing service or a native English speaker with appropriate scientific background.

Comment: In several places the text is over-generalized, for example Line 35 which refers to "aquatic biological composition" – in fact, this study is only about ostracods.

Answer: We agree and will modify the text accordingly to prevent over-generalization. We now refer only to freshwater ostracods throughout the manuscript and exclude the term aquatic biological composition. Ostracodes, however, are recognized as bioindicators of aquatic ecosystems and ecological interactions change. The group belongs to the base of trophic chain and changes on its composition is evidence of aquatic biological composition alteration. The same apply on interpretations of past environments, because ostracodes have abundant fossil record and they reveal changes in the biological composition in response of external variables. We will also modify the title.

Comment: One area that requires clarification is the enumeratation of ostracods (Line 204). Apparently the investigators counted ostracods, but the details of counting aren't presented. How many per sample?

Answer: For this study, we counted specimens in a standard sample volume of 50ml, obtained from filtering 200 L of water in the littoral zones. Then, we used relative abundance for statistical analyses. All this information will be described in detail in the manuscript. Count data expressed as relative abundance is provided in supplementary material https://doi.pangaea.de/10.1594/PANGAEA.940254

Comment:  It's also not clear what they mean by "identified three adults" passage.

Answer: We used three well-preserved adult specimens for microdissection and an accurate identification down to species level. As the target specimens are microcrustaceans, identification based on a single specimen can be misleading, identification based on at least three specimens are considered reliable. We will clarify this sentence in the manuscript.

Comment: Analysis seems to be based on presence/absence, but if count data are available for each sample, then ordination can be based on square root-transformed percentage data.

Answer: Species assemblages were analyzed based on presence/absence data for the NMDS analysis. Because species composition from different sites (latitudes) are evaluated and most species are not shared between lakes, values of species richness and abundances can be highly variable between lakes (highly dissimilar). The NMDS outcome largely depends on the similarity or distance-base index used, therefore, literature recommend using presence/absence data in cases when dissimilarity is relevant.  Please see Ecology, 84(3), 2003, pp. 777–790 by the Ecological Society of America. For the CCA, both presence/absence and relative abundance data resulted in very similar ordination. In the first version of the manuscript, we present the CCA graph of presence/absence data to be homogeneous with NMDS analysis (both with Presence/absence data), but following the view of the reviewer, we will present now the CCA graph based on count data.

Comment: A more serious shortcoming is data handling. Apparently the investigators did not test each variable for normality. Skewed data should be transformed to produce a more normal distribution prior to ordination. For a clear methodology I recommend the most-recent edition of Tabachnick and Fidell's book Using multivariate statistics.

Answer: Environmental data, given different units of quantification, were standardized prior to analysis by subtracting the mean value and dividing by standard deviation. Then, normality was assessed for each variable. Ordination and statistical analysis were therefore performed in a pre-processed database with data normal distributed. As this process may be unclear in the manuscript, we will describe it in detail. We appreciate the recommended literature.

Comment: The authors should be careful in discussing diversity, which is not the same as richness. Also, such comparisons are difficult to interpret because collection size varies among the studies that they cite. Using 'alpha diversity' and 'beta diversity' would help.

Answer: We used Alpha diversity for SEM analysis and following the view of reviewer we will use it to compare our results with other studies conducted worldwide.

Comment:  I suspect that the authors are correct, that geology (local bedrock, karst vs volcanic) is the main influence on ostracod distribution. It's not clear why they use the term 'geodiversity' when bedrock geology alone seems to be the main driver. Data on 'geodiversity', i.e. local diversity of geology (geology, geomorphology, hydrology) aren't presented for each of the lakes.

Answer: The influence of geodiversity on ostracode species distribution is most likely occurring at different hierarchical levels. For instance, we did not find evidence (in SEM analysis) that local bedrock alone explains the species distribution. Currently, the SEM analysis is under review, and we will include local bedrock as an explanatory variable as well. The results of this hypothesis will be discussed in the manuscript. We use the term "geodiversity" because we are providing site characteristics, such as elevation and bedrock type, which constitute part of the geodiversity. We do not provide detailed data on geomorphology and hydrology because this is beyond the scope of the study. We will evaluate the relevance the term "geodiversity" for the manuscript and based on the outcome of SEM analysis, we will define the main driver of limnology and species composition. Then, the title will be change accordingly.

Comment: Another driver seems to be understated, namely precipitation rates. Region YG is low rainfall, Region GSHN is generally higher rainfall. This correlates with karst/low elevation vs volcanic/highlands, but isn't rejected as a main driver. In order to conclude firmly that it's geology rather than rainfall, the authors should add another variable to their data set (i.e. mean annual rainfall at each lake).

Answer: The influences of annual precipitation, precipitation seasonality, air temperature seasonality on species composition was also evaluated. This data was extracted from the Worldclim data base. Precipitation variables were, however, highly correlated with major ions and cations, particularly Mg and Ca. We decided to use the latter for further statistical analysis, as their variability in lakes may explain more clearly and directly ostracode distributions. We will provide a detailed description of variable correlation in the new version of the supplementary material.

 Specific comments –

Answer: We appreciate all these comments and have started to revise the manuscript.

---

## Author Comment (AC3)

Biogeosciences Discuss., author comment AC1
https://doi.org/10.5194/bg-2021-298-AC1, 2022

[Figure]

**Reply on RC1**

Laura Anahí Macario-González et al.
* * *
Author comment on "Geodiversity primarily shapes large-scale limnology and aquatic species distribution in the northern Neotropics" by Laura Anahí Macario-González et al., Biogeosciences Discuss., https://doi.org/10.5194/bg-2021-298-AC1, 2022
* * *
**Reviewer 1.-** We deeply appreciate all your comments and suggestions to improve the manuscript, they contribute significantly to improve the manuscript. Below you will find answers for each of your comments and questions. In case we were unable to follow them, we provide an explanation.

General comments

- Parts of the manuscript are difficult to understand mainly because its language is often not precise enough, the sentence structure is confusing and the grammar is erroneous. Understandability could be improved by reducing different terms for the same purpose, examples are non-biotic/environment, principle component/dimension, elevation/altitude, orography/topography, limnological (sub)regions/limnological (sub)groups, aquatic systems/aquatic ecosystems/aquatic environments/lakes, etc. Some examples of confusing sentence structures and grammatical errors are given in the specific comment section; however, many more issues will remain unlisted and should be edited by the authors, potentially with the help of a native speaker or proofreading service.

Answer: The relevance and usage of the terms in the manuscript is under review. Those terms that cause confusion are being changed and homogenized to improve readability. Sentence structure and grammatical errors will be corrected by a professional English editing service or by a native English speaker with appropriate scientific background.

- The next issue is partly a consequence of the previous issue but not solely: The way how the SEM was set up and how the authors derived their final model is diffuse and not well explained. The parameters which are used to describe geodiversity and limnology, the choice of exogenous and endogenous variables as well as some relationships described by the SEM are not intuitive. For example, (i) in model 2 (Sect S1 in the Supplement), geodiversity (latent variable) is described only by elevation and latitude (observed variables), despite available parameters about the type of bedrock and the mineral composition of lake sediments; (ii) the usage of altitude as endogenous variable (i.e. variables that are dependent in at least one equation), (iii) certain paths

  in the SEM such as the effect of geodiversity on altitude, the effect of limnology on conductivity, although conductivity was used as parameter describing limnology throughout the manuscript, the effect of conductivity on species diversity but not on species associations and the effect of altitude on species associations but not on diversity (see Fig. 6, further obscurities are addressed in the specific comment section). To

improve SEM sections in the manuscript, it needs (i) a more consistent usage of terms normally used in the context of SEM (i.e. exogenous, endogenous, latent, observed variables), (ii) a rework of Fig. 6 which should include observed variables describing latent variables (geodiversity and limnology), (iii) a construction of paths in the SEM based on clearly stated hypothesis and (iv) a more straightforward selection of observed variables to describe latent variables (limnology, geodiversity). The latter could be achieved for example by using environmental variables which were forward selected in the CCA, which was used to examine effects of environment on species composition.

Answer: SEM analysis is under critical review and re-description of the rationale behind variable selection is being conducted with a detailed explanation of the observed variables describing latent variables, as well as exogeneous-endogenous variable selection. We are following the four directions suggested by the reviewer and therefore new paths are expected for this analysis. We did answer questions related to SEM, but detail discussions are not provided here. Detail discussion of the reviewer questions will be included in the manuscript, once the SEM analysis are performed. We, however, expect not show too different results because as observed by another complementary analysis such as CCA, geodiversity is the most important variable explaining limnological patterns and species distribution. The output of all models (not only selected ones, now described in supplementary Material), will be presented in an overview table to complement SEM information.

**Specific comments**

**Abstract**

32: What exactly do you mean with the term "biological composition"? In Fig 6 you use the terms species diversity and species associations. According to Fig 6., the effect of limnology on species associations is not significant, the direct effect of elevation (in Fig 6 called altitude) on species diversity is not tested.

Answer: The term biological composition is currently under review because this introduces confusion to the reader, but in this context, is referring to both species associations (qualitative) and species diversity (quantitative). We will add definitions of the terms used, particularly those related to biogeography, such as species associations, group of species, spatially distinct groups and different habitats. Limnology is not significant for species, except when their indirect influence is evaluated. Elevation is also directly tested and results are relevant for species associations. Please see comment of the SEM analysis in the general comments section.

33: From which result do you derive that geodiversity is the most important driver? Geodiversity is a fundamental driver because it shapes limnology. Hence, I consider geodiversity on a different hierarchical level and difficult to compare to limnology (exogenous vs endogenous variable).

Answer: Here, we assumed that limnology and elevation (with its indirect effect on species diversity) are a function of geodiversity, and given that they better explain the ostracode association, they are assumed as most important drivers. However, following the view of the reviewer, we are considering the effect of geodiversity at different hierarchical level such as the individual influence over limnology and species and a re-discussion of the results are being conducted.

**Introduction**

45 is difficult to understand: What do you mean with "biodiversity is dynamic" and "faster rates"; faster than what?

Answer: We clarified the text and modified as follows:" Biological diversity, defined as the variety of life forms on a place on Earth, is strongly related with geodiversity, as species are

distributed in response to landscape features. Biological diversity is dynamic as species evolve and change distribution patterns at different pace than geodiversity changes"

61: it is unclear what you mean with "biological structure"

Answer: To avoid confusion, we changed here the term "biological structure" by "biogeographical patterns". All terms used to refer to biological systems were homogenized.

71: Confusing sentence structure

Answer: corrected, all manuscript will be revised by a native English speaker

80: Although the study is very comprehensive, it would greatly benefit from an additional layer consisting of data about land use/human activity

Answer: We consider that this information can greatly benefit our study, at the moment we do not have such information for all our study sites. Evaluate land use/human activities is beyond of the scope of this study, as we analyze the relationships between geology (geodiversity), limnology and species composition. Indirectly, we analyze the effect of human impacts on aquatic ecosystems using secchi depth, and other trophy state variables. In the conclusions, we will add a paragraph of future directions for this study, and particularly describing the importance of the influence of land use and human activity on aquatic ecosystems and biological change.

88: Confusing sentence structure

Answer: corrected, same comment than in line 71.

94: ostracods are a well-suited group

Answer: corrected according to the reviewer suggestion

94: The study is not investigating traits.

Answer: following the view of the reviewer we modify the term and use "topic"

94: singular: ostracod, plural: ostracods. Rephrased sentence: Ostracods are bivalved microcrustaceans which are abundant, diverse and widely distributed in recent environments.

Answer: changed according to reviewer suggestion

103: You are also looking at effects on species diversity.

In general, terms like diversity, composition, assemblages, associations, species distribution, biological structure, are not well defined and often used synonymously. To avoid confusion, please stick to the same expression throughout the manuscript if the purpose is the same.

Answer: We carefully verify all terms and those used inappropriate were changed and homogenized to avoid confusion. We will also provide a clear definition for species associations and other biogeographic terms used.

**Material and Methods**

156: Here you use "non-biological" and in other parts "environmental", I suggest to stick to either "environmental" or "abiotic" throughout the entire manuscript, also in figures.

Answer: We used "environmental" instead of "non-biological", throughout the manuscript and figures

186: How did you handle missing data?

Answer: for statistical purposes, missing data were completed with average values of the respective variable. This information is included in the manuscript.

200: The maps do either not represent the measured data or it is not visible. Please add the measured data. Also, a reference to the figure is missing.

Answer: Maps were re-designed and measured data were integrated to the interpolation map to clearly visualize the power of prediction. We also added the reference for the figure.

218: Clarify how you distinguished species groups. Was it manually done by visually examining the graph?

Answer: For species group determination we apply a hierarchical cluster analysis based on Ward distances and then overlapped (hclust in R) on the NMDS ordination. These techniques usually validate one to another and reduce the uncertainty for group boundaries determination. We describe this procedure in the manuscript.

221: Here you use "relating non-biological variables" and later in the paragraph "environmental variables", take care of consistency.

Answer: corrected throughout the text, we used environmental variable.

221: Besides geological effects, you also assessed limnological effects (temperature, conductivity, etc.)

Answer: corrected, we included a paragraph on the limnological effect

231: What do you mean with "related environmental variables"?

Answer: We are referring to the variables that derived or are influenced by geology, namely, sediment geochemistry and elevation. To clarify the sentence, we excluded the phrase "related environmental variables" and those considered in the analysis were enlisted.

231: You also assessed the influence of geodiversity on species diversity not only on the composition (in Fig 6 called "species association").

Answer: This is correct, we included it in the text to clarify methods

235: It is not clear that you use elevation gradients, latitude and bedrock as observed variables to explain geodiversity (latent variable). The same applies for limnology and its observed variables.

Answer: Please see the answer to general comments.

236: It is not clear if geodiversity is assumed as indirect, direct or both, the same for limnology

Answer: Geodiversity is in our analysis assumed as a direct effect on limnology and indirect to species. The influence of limnology on species is evaluated as a direct influence.

237: It is not clear, how you take vulcanism, precipitation and marine-freshwater interactions into account and where the major anion and cations belong to in the SEM.

Answer: Please see general comments for SEM

**Results**

Results are normally written in simple past tense

Answer: The text is currently under revision by native American scientist. We will try to have this issue corrected.

253: Here and in Table 1 the term "(sub)groups" is used, in the text mainly "sub(regions)". In general, I think the terms "limnological classification" and "limnological regions" are confusing as you also use the term "limnology" as hypernym for water chemicals and physical properties of the aquatic ecosystems. The "limnological classification", however, is based next to limnological variables also on geological, sedimentological and mineralogical variables.

Answer: We homogenized the term and use sub-regions, the term sub-groups was deleted to avoid confusion. Also, the term limnological classification was changed to geolimnological classification to be more precise and not to be confused with limnological variables.

Fig 2 (c): It is difficult to track dots to site abbreviations. Also, site 65 appears two times once with the site abbreviation CHI and once with CH1

Answer: we try to clarify and better link dots with site abbreviations, however, because of the image size, we refer to figure 1 to identify site abbreviations. The site 65 which was duplicated, will be corrected.

271: Are the "thirteen variables" those which were selected based on "superimposed arrows"? Please clarify

Answer: Yes, all statistical analyses were conducted using a data base with uncorrelated variables, those represented by superimposed arrows in the PCA ordination were deleted, as they demonstrated similar response. This procedure was described in detail in methodology and in the results sections.

Fig 3 and 4 (b-d): Could you show sites with colours according to the observed values to see how well they fit into the predicted surface. Write the variables which are mapped in the graphs also in the legend or put them as title.

Answer: We added measured values to the predicted map to verify the prediction power of the algorithm. We also added the variable name on the legend and as title to facilitate the visual recognition.

300: You write about "sedimentology and geology" as important variables. However, there are no variables called like this. In order to make that point clear, I think it would help, if parameters in Fig 3 (a) and Fig 4 (a) are coloured according to their type (i.e. limnology, sedimentology, geology, mineralogy). This would also help to not confound carbonate measurements derived from the water with measurements derived from the sediment.

Answer: We appreciate this observation; we conducted the recommendation and figure 3 (a) and 4 (a) were modified.

358: "supporting NMDS ordination" or supporting group selection?

Answer: supporting group selection is correct; the text was changed accordingly.

361: Are you deriving the tolerance to high conductivity from the literature or from your CCA? If the latter is true, you should refer to Fig. S3.

Answer: We derived high conductivity tolerance from the results of our analysis, and we refer to figure S3.

397: In S1 you mention five models instead of six.

Answer: We corrected accordingly

Also, in S1 you write "… we assume that **elevation gradients**, **bedrock** and **latitude** were primary factors determining biological composition in aquatic systems. These three factors were then used as exogenous variables…" Here you state that initially **geodiversity**, **limnology** and **elevation** were your three exogenous variables. It is not clear which variables describe geodiversity and limnology.

Answer: This is a terminology issue, latitude was associated to limnology because of ionic composition of waters (expected to be affected by precipitation); bedrock was associated to geodiversity. Terminology issues are corrected to exclude such confusion

408: Why do you test the influence of species associations on diversity? What is the hypothesis?

Answer: The analysis intends to evaluate the effect of the limnology over diversity (numerical data) and species associations (no numerical data). Relationship between diversity and species composition are addressed only to estimate indirect effect of geodiversity.

408: Why "indirect" when there is a "direct" link from limnology to species associations?

Answer: Here we consider that the confusion is an issue of the wide variety of terms used in the text. In this case, we changed to direct effect.

Fig. 6: instead adding "environmental variables" to limnology, add observed variables which describe limnology, the same applies for geodiversity. Why are you not looking at the direct effect of limnology on species diversity?

Answer: Please see our answer to general comments on SEM

**S1**

In general, it is a good idea to provide details about all SEMs, however, the text in S1 often explains the same as the main text, but in a different way, which adds to the confusion.

Paragraph 3: "Geodiversity was constructed only with elevation and latitude as predictors, whereas limnology only with conductivity. The selection of these variables resulted from the fact that elevation was directly related with water temperature in lakes and latitude with presence of carbonates given reduction in precipitation and increase evaporation" I don't understand why elevation and latitude were used as observed variables for geodiversity. First of all, if elevation was related to water temperature and latitude to the presence of carbonates, why not take water temperature and carbonates as observed variables, instead of related variables. But secondly, water temperature was always part of the variables describing limnology and not geodiversity.

Paragraph 4: Model 4 "was constructed on the basis of the model 2 and 3 with respect of predictors of geodiversity and limnology". It is not clear, which observed variables are actually used for geodiversity and limnology.

Answer: We appreciate all these questions related to the model, a critical analysis is being conducted taking into account all of them. Please see general comments to the SEM.

**Discussion**

419: What do you mean with "Geology and associated variables"?

Answer: "associated variables" in this case, are those which are directly influenced by geology such as sediment composition and elevation. In order to avoid confusion, we excluded the term "associated variables" and clearly describe them.

493: Your results show a different picture, see line 277: "pH was highly correlated (>0.73) with the second component (PC2), suggesting that it is the second most influential variable of the YG aquatic environments (Fig. 3a, Table S2.1)"

Answer: we consider that the text is congruent with the figure, however, we found the term "second most influential variable" confusing, and we will modify the text accordingly.

Chapter 4.1. is well written

511: I suggest to state this more carefully as you are only looking at a handful of lakes without applying any statistical analysis to test this pattern.

Answer: We agree that we are over-generalizing some of our interpretations such as in this case. We will carefully check this to avoid inaccurate assumptions. We consider that pointing out what the results are covering will very much improve the manuscript. For example, instead of using aquatic communities, we will use ostracods (the target group).

539: The obvious spatial pattern of species composition may also be a hint to dispersal related processes which are not at all touched in this study. A potential statistical way to incorporate spatiality in this study would be to include space as exogenous latent variable in the SEM with latitude and longitude as observed variables. The possibility of dispersal limitations acting as additional driver structuring ostracod communities should at least be discussed.

Answer: We appreciate this comment, and the topics suggested will be included in our SEM analysis.

546: What is the indirect effect of limnology on species composition? In Fig 6. I see a direct effect of limnology on species associations (I guess this is meant with species composition), and a questionable (see other comments) indirect effect via conductivity on species diversity.

556: "elevation" is not used as a variable in the CCA

Answer: elevation is a driver of temperature, the term is corrected.

575: "as evolutionary trait"?

Answer: This issue is related with the grammatical errors conducted throughout the study, but inaccurate terms are being corrected.

**Technical comments**

Answer: All technical comments are considered and included in the manuscript.

---

## Author Response (AR1)

Biogeosciences Discuss., author comment AC1
https://doi.org/10.5194/bg-2021-298-AC1, 2022

[Figure]

**Reply on RC1**

Laura Anahí Macario-González et al.
* * *
Author comment on "Geodiversity influences limnological characteristics and freshwater ostracode species distributions across broad spatial scales in the northern Neotropics" by Laura Anahí Macario-González et al., Biogeosciences, https://doi.org/10.5194/bg-2021-298-AC1, 2022
* * *
**Reviewer 1.-** We deeply appreciate all your comments and suggestions to improve the manuscript, they contribute significantly to improve the manuscript. Below you will find answers for each of your comments and questions. In case we were unable to follow them, we provide an explanation.

General comments

- Parts of the manuscript are difficult to understand mainly because its language is often not precise enough, the sentence structure is confusing and the grammar is erroneous. Understandability could be improved by reducing different terms for the same purpose, examples are non-biotic/environment, principle component/dimension, elevation/altitude, orography/topography, limnological (sub)regions/limnological (sub)groups, aquatic systems/aquatic ecosystems/aquatic environments/lakes, etc. Some examples of confusing sentence structures and grammatical errors are given in the specific comment section; however, many more issues will remain unlisted and should be edited by the authors, potentially with the help of a native speaker or proofreading service.

Answer: The relevance and usage of the terms in the manuscript was reviewed. Those terms that cause confusion are changed and homogenized to improve readability. Sentence structure and grammatical errors are corrected by a native English speaker with appropriate scientific background.

- The next issue is partly a consequence of the previous issue but not solely: The way how the SEM was set up and how the authors derived their final model is diffuse and not well explained. The parameters which are used to describe geodiversity and limnology, the choice of exogenous and endogenous variables as well as some relationships described by the SEM are not intuitive. For example, (i) in model 2 (Sect S1 in the Supplement), geodiversity (latent variable) is described only by elevation and latitude (observed variables), despite available parameters about the type of bedrock and the mineral composition of lake sediments; (ii) the usage of altitude as endogenous variable (i.e. variables that are dependent in at least one equation), (iii) certain paths

in the SEM such as the effect of geodiversity on altitude, the effect of limnology on conductivity, although conductivity was used as parameter describing limnology throughout the manuscript, the effect of conductivity on species diversity but not on species associations and the effect of altitude on species associations but not on diversity

(see Fig. 6, further obscurities are addressed in the specific comment section). To improve SEM sections in the manuscript, it needs (i) a more consistent usage of terms normally used in the context of SEM (i.e. exogenous, endogenous, latent, observed variables), (ii) a rework of Fig. 6 which should include observed variables describing latent variables (geodiversity and limnology), (iii) a construction of paths in the SEM based on clearly stated hypothesis and (iv) a more straightforward selection of observed variables to describe latent variables (limnology, geodiversity). The latter could be achieved for example by using environmental variables which were forward selected in the CCA, which was used to examine effects of environment on species composition.

Answer: SEM analysis was reviewed and re-description of the rationale behind variable selection was conducted with a detailed explanation of the observed variables describing latent variables, as well as exogeneous-endogenous variable selection. We followed the four directions suggested by the reviewer and therefore new paths are explained. Detail discussion of the reviewer questions is included in both results section of the manuscript and section 1 of the supplementary material. Geodiversity resulted the most important variable explaining limnological patterns and in turn, limnology to species composition (distribution). The metrics of fit of all models are presented in supplementary material.

**Specific comments**

**Abstract**

32: What exactly do you mean with the term "biological composition"? In Fig 6 you use the terms species diversity and species associations. According to Fig 6., the effect of limnology on species associations is not significant, the direct effect of elevation (in Fig 6 called altitude) on species diversity is not tested.

Answer: The term biological composition was reviewed because it introduces confusion to the reader. We add definitions of the terms used, particularly geodiversity, biological composition and limnology. Limnology is not significant for species, except when their indirect influence is evaluated. Elevation is also directly tested and results are relevant for species associations.

33: From which result do you derive that geodiversity is the most important driver? Geodiversity is a fundamental driver because it shapes limnology. Hence, I consider geodiversity on a different hierarchical level and difficult to compare to limnology (exogenous vs endogenous variable).

Answer: The PCA and the SEM reveal that geolimnology is the most important variable differentiating limnological regions. Here, we assumed that limnology and elevation (with its indirect effect on species diversity) are a function of geodiversity, and given that they better explain the ostracode association, they are assumed as most important drivers. However, following the view of the reviewer, we are considering the effect of geodiversity at different hierarchical level such as the individual influence over limnology and species and we re-discus the results.

**Introduction**

45 is difficult to understand: What do you mean with "biodiversity is dynamic" and "faster rates"; faster than what?

Answer: We clarified the text and modified as follows: "Biological diversity, defined as the variety of life forms in a place on Earth (Huston, 1995), is strongly related to geodiversity, as species are distributed in response to landscape features. Biological diversity is dynamic. Species evolve and distribution patterns change at rates different from rates of change of geodiversity"

61: it is unclear what you mean with "biological structure"

Answer: To avoid confusion, we changed here the term "biological structure" by "biogeographical patterns". All terms used to refer to biological systems were homogenized.

71: Confusing sentence structure

Answer: corrected, all manuscript will be revised by a native English speaker

80: Although the study is very comprehensive, it would greatly benefit from an additional layer consisting of data about land use/human activity

Answer: We consider that this information can greatly benefit our study, now we do not have land use data for all our study sites. Evaluate land use/human activities is beyond of the scope of this study, as we analyze the relationships between geology (geodiversity), limnology and species composition. Indirectly, we analyze the effect of human impacts on aquatic ecosystems using secchi depth, and other trophy state variables. In the conclusions, we add a paragraph of future directions for this study, and particularly describing the importance of the influence of land use and human activity on aquatic ecosystems and biological change.

88: Confusing sentence structure

Answer: corrected, same comment than in line 71.

94: ostracods are a well-suited group

Answer: corrected according to the reviewer suggestion

94: The study is not investigating traits.

Answer: following the view of the reviewer we modify the term

94: singular: ostracod, plural: ostracods. Rephrased sentence: Ostracods are bivalved microcrustaceans which are abundant, diverse and widely distributed in recent environments.

Answer: changed according to reviewer suggestion

103: You are also looking at effects on species diversity.

In general, terms like diversity, composition, assemblages, associations, species distribution, biological structure, are not well defined and often used synonymously. To avoid confusion, please stick to the same expression throughout the manuscript if the purpose is the same.

Answer: We carefully verify all terms and those used inappropriate were changed and homogenized to avoid confusion. We provide a clear definition for biogeographic terms used.

**Material and Methods**

156: Here you use "non-biological" and in other parts "environmental", I suggest to stick to either "environmental" or "abiotic" throughout the entire manuscript, also in figures.

Answer: We used "environmental" instead of "non-biological", throughout the manuscript and figures

186: How did you handle missing data?

Answer: for statistical purposes, missing data were completed with average values of the respective variable, as missing data represented less than 8% of the dataset. This information is included in the manuscript.

200: The maps do either not represent the measured data or it is not visible. Please add the measured data. Also, a reference to the figure is missing.

Answer: Maps were re-designed and measured data were integrated to the interpolation map to clearly visualize the power of prediction. We also added the reference for the figure.

218: Clarify how you distinguished species groups. Was it manually done by visually examining the graph?

Answer: For species group determination we apply a hierarchical cluster analysis based on Ward distances and then overlapped (hclust in R) on the NMDS ordination. These techniques usually validate one to another and reduce the uncertainty for group boundaries determination. We describe this procedure in the manuscript.

221: Here you use "relating non-biological variables" and later in the paragraph "environmental variables", take care of consistency.

Answer: corrected throughout the text, we used environmental variable.

221: Besides geological effects, you also assessed limnological effects (temperature, conductivity, etc.)

Answer: corrected, we included a paragraph on the limnological effect

231: What do you mean with "related environmental variables"?

Answer: We are referring to the variables that derived or are influenced by geology, namely, sediment geochemistry and elevation. To clarify the sentence, we excluded the phrase "related environmental variables" and those considered in the analysis were enlisted.

231: You also assessed the influence of geodiversity on species diversity not only on the composition (in Fig 6 called "species association").

Answer: This is correct, we included it in the text to clarify methods

235: It is not clear that you use elevation gradients, latitude and bedrock as observed variables to explain geodiversity (latent variable). The same applies for limnology and its observed variables.

Answer: Please see the answer to general comments.

236: It is not clear if geodiversity is assumed as indirect, direct or both, the same for limnology

Answer: Geodiversity and limnology was evaluated as both exogenous variables and their direct and indirect influence on species composition and richness was tested. Please see section 1 of supplementary material.

237: It is not clear, how you take vulcanism, precipitation and marine-freshwater

interactions into account and where the major anion and cations belong to in the SEM.

Answer: Please see general comments for SEM

**Results**

Results are normally written in simple past tense

Answer: The text was reviewed by native American scientist.

253: Here and in Table 1 the term "(sub)groups" is used, in the text mainly

"sub(regions)". In general, I think the terms "limnological classification" and "limnological regions" are confusing as you also use the term "limnology" as hypernym for water chemicals and physical properties of the aquatic ecosystems. The "limnological classification", however, is based next to limnological variables also on geological, sedimentological and mineralogical variables.

Answer: We homogenized the term and use sub-regions, the term sub-groups was deleted to avoid confusion. Also, the term limnological was defined to be more precise.

Fig 2 (c): It is difficult to track dots to site abbreviations. Also, site 65 appears two times once with the site abbreviation CHI and once with CH1

Answer: we clarify and better link dots with site abbreviations, however, because of the image size, we refer to figure 1 to identify site abbreviations. The site 65 which was duplicated is corrected.

271: Are the "thirteen variables" those which were selected based on "superimposed arrows"? Please clarify

Answer: Yes, all statistical analyses were conducted using a data base with uncorrelated variables, those represented by superimposed arrows in the PCA ordination were deleted, as they demonstrated similar response. This procedure was described in detail in methodology and in the results sections.

Fig 3 and 4 (b-d): Could you show sites with colours according to the observed values to see how well they fit into the predicted surface. Write the variables which are mapped in the graphs also in the legend or put them as title.

Answer: We added measured values to the predicted map to verify the prediction power of the algorithm. We also added the variable name on the legend and as title to facilitate the visual recognition.

300: You write about "sedimentology and geology" as important variables. However, there are no variables called like this. In order to make that point clear, I think it would help, if parameters in Fig 3 (a) and Fig 4 (a) are coloured according to their type (i.e. limnology, sedimentology, geology, mineralogy). This would also help to not confound carbonate measurements derived from the water with measurements derived from the sediment.

Answer: We appreciate this observation; we conducted the recommendation and figure 3 (a) and 4 (a) were modified.

358: "supporting NMDS ordination" or supporting group selection?

Answer: supporting group selection is correct; the text was changed accordingly.

361: Are you deriving the tolerance to high conductivity from the literature or from your CCA? If the latter is true, you should refer to Fig. S3.

Answer: We derived high conductivity tolerance from the results of our analysis, and we refer to figure S3.

397: In S1 you mention five models instead of six.

Answer: We corrected accordingly

Also, in S1 you write "… we assume that **elevation gradients**, **bedrock** and **latitude** were primary factors determining biological composition in aquatic systems. These three factors were then used as exogenous variables…" Here you state that initially **geodiversity**, **limnology** and **elevation** were your three exogenous variables. It is not clear which variables describe geodiversity and limnology.

Answer: This is a terminology issue, latitude was associated to limnology because of ionic composition of waters (expected to be affected by precipitation); bedrock was associated to geodiversity. Terminology issues are corrected to exclude such confusion

408: Why do you test the influence of species associations on diversity? What is the hypothesis?

Answer: The analysis of species association on diversity was excluded.

408: Why "indirect" when there is a "direct" link from limnology to species associations?

Answer: Here we consider that the confusion is an issue of the wide variety of terms used in the text. In this case, we changed to direct effect.

Fig. 6: instead adding "environmental variables" to limnology, add observed variables which describe limnology, the same applies for geodiversity. Why are you not looking at the direct effect of limnology on species diversity?

Answer: Please see our answer to general comments on SEM

**S1**

In general, it is a good idea to provide details about all SEMs, however, the text in S1 often explains the same as the main text, but in a different way, which adds to the confusion.

Paragraph 3: "Geodiversity was constructed only with elevation and latitude as predictors, whereas limnology only with conductivity. The selection of these variables resulted from the fact that elevation was directly related with water temperature in lakes and latitude with presence of carbonates given reduction in precipitation and increase evaporation" I don't understand why elevation and latitude were used as observed variables for geodiversity. First of all, if elevation was related to water temperature and latitude to the presence of carbonates, why not take water temperature and carbonates as observed variables, instead of related variables. But secondly, water temperature was always part of the variables describing limnology and not geodiversity.

Paragraph 4: Model 4 "was constructed on the basis of the model 2 and 3 with respect of predictors of geodiversity and limnology". It is not clear, which observed variables are actually used for geodiversity and limnology.

Answer: We appreciate all these questions related to the model design, a critical analysis was conducted taking into account all of them. Please see general comments to the SEM.

**Discussion**

419: What do you mean with "Geology and associated variables"?

Answer: "associated variables" in this case, are those which are directly influenced by geology such as mineralogy, sediment geochemistry and elevation. In order to avoid confusion, we excluded the term "associated variables" and explicitly describe them.

493: Your results show a different picture, see line 277: "pH was highly correlated (>0.73) with the second component (PC2), suggesting that it is the second most influential variable of the YG aquatic environments (Fig. 3a, Table S2.1)"

Answer: we consider that the text is congruent with the figure, however, we found the term "second most influential variable" confusing, and we will modify the text accordingly.

Chapter 4.1. is well written

511: I suggest to state this more carefully as you are only looking at a handful of lakes without applying any statistical analysis to test this pattern.

Answer: We agree that we are over-generalizing some of our interpretations such as in this case. We will carefully check this to avoid inaccurate assumptions. We consider that pointing out what the results are covering will very much improve the manuscript. For example, instead of using aquatic communities, we will use ostracods (the target group).

539: The obvious spatial pattern of species composition may also be a hint to dispersal related processes which are not at all touched in this study. A potential statistical way to incorporate spatiality in this study would be to include space as exogenous latent variable in the SEM with latitude and longitude as observed variables. The possibility of dispersal limitations acting as additional driver structuring ostracod communities should at least be discussed.

Answer: We appreciate this comment, and the topics suggested was included in our SEM analysis.

546: What is the indirect effect of limnology on species composition? In Fig 6. I see a direct effect of limnology on species associations (I guess this is meant with species composition), and a questionable (see other comments) indirect effect via conductivity on species diversity.

556: "elevation" is not used as a variable in the CCA

Answer: elevation is a driver of temperature, the term is corrected.

575: "as evolutionary trait"?

Answer: This issue is related with the grammatical errors conducted throughout the study, but inaccurate terms are being corrected.

**Technical comments**

Answer: All technical comments are considered and included in the manuscript.

[Figure]

Biogeosciences Discuss., author comment AC1
https://doi.org/10.5194/bg-2021-298-AC1, 2022 ©
Author(s) 2022. This work is distributed under the
Creative Commons Attribution 4.0 License.

[Figure]

**Reply on RC2**

Laura Anahí Macario-González et al.
* * *
Author comment on " Geodiversity influences limnological characteristics and freshwater ostracode species distributions across broad spatial scales in the northern Neotropics" by Laura Anahí Macario-González et al., Biogeosciences, https://doi.org/10.5194/bg-2021-298-AC1, 2022
* * *
**Reviewer 2.-** We appreciate all the comments and suggestions, they significantly improve the manuscript. Below we provide answers to all comments. We have tried to address most of the suggestions made by reviewer 2, but in case this is not possible, we provide a detailed explanation.

Comment: Because the authors are evidently not native speakers of English, there are considerable numbers of grammatical mistakes and poorly-phrased passages, most of which are detailed below with suggestions for improvement.

Answer: Sentence structure and grammatical errors were corrected by a native English speaker with appropriate scientific background.

Comment: In several places the text is over-generalized, for example Line 35 which refers to "aquatic biological composition" – in fact, this study is only about ostracods.

Answer: We agree and modified the text accordingly to prevent over-generalization. We now refer only to freshwater ostracods throughout the manuscript and exclude the term aquatic biological composition. Ostracodes, however, are recognized as bioindicators of aquatic ecosystems and ecological interactions change. The group belongs to the base of trophic chain and changes on its composition is evidence of aquatic biological composition alteration. The same apply on interpretations of past environments, because ostracodes have abundant fossil record and they reveal changes in the biological composition in response of external variables. We also modify the title.

Comment: One area that requires clarification is the enumeratation of ostracods (Line 204). Apparently the investigators counted ostracods, but the details of counting aren't presented. How many per sample?

Answer: For this study, we counted specimens in a standard sample volume of 50ml, obtained from filtering 200 L of water in the littoral zones. Then, we used relative abundance for statistical analyses. All this information is described in detail in the manuscript. Count data expressed as relative abundance is provided in supplementary material https://doi.pangaea.de/10.1594/PANGAEA.940254

Comment: It's also not clear what they mean by "identified three adults" passage.

Answer: We used three well-preserved adult specimens for microdissection and an accurate identification down to species level. As the target specimens are microcrustaceans, identification based on a single specimen can be misleading because of morphological plasticity and mutations, identification based on at least three specimens are considered reliable. We clarified this sentence in the manuscript.

Comment: Analysis seems to be based on presence/absence, but if count data are available for each sample, then ordination can be based on square root-transformed percentage data.

Answer: Species assemblages were analyzed based on presence/absence data for the NMDS analysis. Because species composition from different sites (latitudes) are evaluated and most species are not shared between lakes, values of species richness and abundances are highly variable between lakes (highly dissimilar). The NMDS outcome largely depends on the distance-base index used, therefore, literature recommend using presence/absence data in cases when dissimilarity is relevant. Please see Ecology, 84(3), 2003, pp. 777–790 by the Ecological Society of America. For the CCA, both presence/absence and relative abundance data resulted in very similar ordination. In the first version of the manuscript, we present the CCA graph of presence/absence data to be homogeneous with NMDS analysis (both with Presence/absence data), but following the view of the reviewer, we present now the CCA graph based on count data.

Comment: A more serious shortcoming is data handling. Apparently the investigators did not test each variable for normality. Skewed data should be transformed to produce a more normal distribution prior to ordination. For a clear methodology I recommend the most-recent edition of Tabachnick and Fidell's book Using multivariate statistics.

Answer: Environmental data, given different units of quantification, were standardized prior to analysis by subtracting the mean value and dividing by standard deviation. Then, normality was assessed for each variable. Ordination and statistical analysis were therefore performed in a pre-processed database with data normal distributed. As this process may be unclear in the manuscript, we described it in detail. We appreciate the recommended literature.

Comment: The authors should be careful in discussing diversity, which is not the same as richness. Also, such comparisons are difficult to interpret because collection size varies among the studies that they cite. Using 'alpha diversity' and 'beta diversity' would help.

Answer: We used alpha diversity to estimate diversity in the region, when possible we used this index to compare our results with other studies conducted worldwide.

Comment: I suspect that the authors are correct, that geology (local bedrock, karst vs volcanic) is the main influence on ostracod distribution. It's not clear why they use the term 'geodiversity' when bedrock geology alone seems to be the main driver. Data on 'geodiversity', i.e. local diversity of geology (geology, geomorphology, hydrology) aren't presented for each of the lakes.

Answer: The influence of geodiversity on ostracode species distribution is most likely occurring at different hierarchical levels. For instance, we did not find evidence (in SEM analysis) that local bedrock alone explains the species distribution. The hypothesis postulated by reviewer was evaluated and discussed in the manuscript. We use the term "geodiversity" because we are providing site characteristics, such as elevation and bedrock type, which constitute part of the geodiversity. We do not provide detailed data on geomorphology and hydrology because this is beyond the scope of the study. We evaluated the relevance of the term "geodiversity" and we decided to continue using it.

Comment: Another driver seems to be understated, namely precipitation rates. Region YG is low rainfall, Region GSHN is generally higher rainfall. This correlates with karst/low elevation vs volcanic/highlands, but isn't rejected as a main driver. In order to conclude firmly that it's geology rather than rainfall, the authors should add another variable to their data set (i.e. mean annual rainfall at each lake).

Answer: The influences of annual precipitation, precipitation seasonality, air temperature seasonality on species composition was exploratory evaluated in the first stage of the study. This data was extracted from the Worldclim data base. Precipitation variables were, however, highly correlated with major ions and cations, particularly Mg and Ca. We decided to use the latter for the analysis, as their variability in lakes may explain more clearly and directly ostracode distributions. We also considered precipitation as a climatic variable which are out of the scope of the manuscript.

 Specific comments –

Answer: We appreciate all these comments and all of them were included to the manuscript.

---

## Referee Report (RR1)

**Second** Review of Article
May 9, 2022

Title: Geodiversity primarily shapes large-scale limnology and aquatic species distribution in the northern Neotropics
Author(s): Laura Anahí Macario-González et al.
MS No.: bg-2021-298 version 3

**General comments**

This is a vastly improved manuscript. There are only two general issues that should be addressed throughout this manuscript; once addressed, I consider the manuscript to be acceptable for publication.

*Geodiversity*. I appreciate the defining of "geodiversity" at Line 38. It seems that the term "geodiversity" is synonymous with "local environmental conditions", a term that is more clear and is used nearly exclusively (at least in the multiple studies that I have seen). It's not clear why "geodiversity" is preferred, and in fact the term is misleading -- especially the "diversity" part of the word (which implies differences among sites, rather than denoting a fixed set of conditions). The authors should at least indicate this synonymy, e.g. near Line 38. Note that this is especially misleading at Line 618. I encourage the authors to *consider* replacing each "geodiversity" with "environmental conditions" (or a variation thereof) throughout, unless there's a convincing reason that "geodiversity" is indeed the better term.

*Sample sizes*. Apparently only six individuals (or three?) were identified at some sites, but as many as 60 were at other sites. A better mention of the numbers identified is essential, for example at Line 217. This creates the *potential* for a substantial error due to sampling effort. For sites with six identified individuals, the percentages are coarse approximations at best. This study is still useful – we all deal with compromised or limited data sets -- but this potential limitation must be addressed in the Discussion.

**Specific comments**

Line 156: Please give the reader an idea of how many samples were *typically* collected. If most lakes were represented by two samples (i.e. 6 individuals) but some were represented by ten samples (i.e. 30 individuals), this may affect the apparent distribution of species as well as the calculated richness and Shannon-Weiner diversity indices.

Line 215: Change "was" to "were".

Line 216: Either transpose "stereomicroscope" and "Leica MZ75", or place "Leica MZ75" in parentheses.

Line 217: It seems that only 3 individuals were identified in each sample, and up to 10 samples were collected at each lake (Line 156); please report the average number of samples per lake. Multiple sampling is appropriate and should be repeated here at Line 217. I'm not familiar with ostracod procedures, but three is a very small number. If this is a reasonable number, please indicate and include references. For example, add "Although a sample size of three is very small, this is consistent with ostracod studies in general (reference, year) in part due to low ostracod abundances in most sediments worldwide (reference, year)." In addition, see comment for Line 379.

Line 244: Delete "the" and transpose "software" and "Canoco version 5".

Line 248: Please specify the software that includes SEM. It may be part of R, but this isn't mentioned until Line 264, so a mention is needed early in this section.

Line 379: Please report the total number of species.

Line 421: I see the groups of ostracods from NMDS is Figure 5(a), but I would very much like to see the eigenvectors from CCA. A figure similar to Figure 4(a), but based on CCA, is important.

Line 423: We're told that local geology is the ultmate driver, and I accept this as true. I think its effect is delivered through evaporative concentration (PCA shows that ionic concentration varies strongly among these systems). In other words, ionic concentration is the immediate cause;

what matters *directly* to ostracods is water conditions, so local geology is an *indirect* factor. A figure with NMDS eigenvectors would be very, very useful for this.

Line 447: Insert comma after "limnology".

Line 460 (Figure 6): I like this figure very much; it's clear yet detailed. I'm somewhat confused, however: doesn't "geodiversity" *include* limnology, conductivity, and elevation? I'm not sure that "NMDS" is needed in the "Species composition" box; please verify and/or put "NMDS" in parentheses.

Line 467: Replace "drives" with "defines".

Line 506: It's not clear that volcanism created most Central American lakes; in my experience, at least as many were formed by mass movement (in higher elevations). At best, the authors could say that "Volcanism is a common mode of formation".

Line 547: Line 217 reports that three ostracods were identified per sample, yet Line 547 reports up to nine species per lake.

Line 551: Collection effort can make a huge difference here. If ten samples were collected from one lake but only two from another, we'd expect greater richness in the 10-sample lake. In addition, two samples may be enough to characterize small lakes (*if* sufficient numbers of ostracods are identified in each sample), but this is less true for larger lakes. At least this should be acknowledged here: "While the number of samples per lake varied from X to Y"….

Line 618: transpose "abundant aquatic systems" and "high geodiversity". If "geodiversity" = "environmental conditions", then this phrase should be "high diversity of environmental conditions" rather than "geodiversity". This is where the "diversity" in "geodiversity" becomes confusing.

---

## Author Response (AR2)

Biogeosciences Discuss., author comment AC1
https://doi.org/10.5194/bg-2021-298-AC1, 2022

[Figure]

**Reply on RC1 (second revision)**

Laura Anahí Macario-González et al.
* * *
Author comment on "Geodiversity influences limnological characteristics and freshwater ostracode species distributions across broad spatial scales in the northern Neotropics" by Laura Anahí Macario-González et al., Biogeosciences, https://doi.org/10.5194/bg-2021-298-AC1, 2022
* * *
**Reviewer 1.-** We deeply appreciate all your comments and suggestions to improve the manuscript. Below you will find answers to each of your comments and questions. In case we were unable to follow them, we provide an explanation.

General comments

Main text

Parts of the methods and most of the results are still written in present tense. Both sections are normally reported in the past tense. Two examples out of many: 187: The final data set includes 23 variables, of which 21 are numerical… 297: The PC1 accounts for 23.4%...

Answer: We carefully check the verb grammatical tense in the "Methodology" and "Result" sections. We additionally check grammatical time of verbs of other sections. Verbs were changed to their correct grammatical time.

204: I think the name of the r package that you used is "PCAmixdata"

Answer: PCAmixdata is the correct name of the package, this was changed throughout the manuscript.

288: "Limnological classification"; In the previous review, I commented "… the terms "limnological classification" and "limnological regions" are confusing as you also use the term "limnology" as hypernym for water chemicals and physical properties of the aquatic ecosystems. The "limnological classification", however, is based next to limnological variables also on geological, sedimentological and mineralogical variables."; and you answered "…the term limnological classification was changed to geolimnological classification to be more precise and not to be confused with limnological variables.". I could not find these changes in the revised version of the manuscript.

Answer: We changed the term, but we decided not to use "geolimnology", as it may produce additional confusion. We stick with the term "limnological regions" used in other sections of the manuscript. The corrected text is as follows: "Limnological regions in the northern Neotropical region, with lakes main water physical, chemical, mineralogical, and geological properties"

305: ad figure 3 (a) and figure 4 (a): The two axes (component 1 and component 2) need to be on the same scale, otherwise arrows which rather follow the direction of component 1 seem more important.

Answer: figures were rescaled to the same extent to avoid graph misinterpretation.

436: Only species composition was included, not richness?

Answer: The analysis was performed on both species composition and richness, thus, we included richness to the text.

450: The explanations for CFI, RMSEA and SRMR need to appear when you first mention these metrics (line 436)

Answer: Following the suggestion of the reviewer, we provide the explanations for global metrics of fit CFI, RMSEA and SRMR in line 436.

600-601: It is not clear, what the values in the brackets mean "(<0.1)", "(>2.0)". Also consider a different word for "insignificant", because the paths are significant, however, of minor importance.

Answer: To avoid confusion, symbols < and > in the brackets were deleted and the significance value was used. Following the recommendation of the reviewer we exclude "insignificant" and used "significant with minor importance"

Supplement

I am not familiar with the output table of SEMs, however, it seems that conductivity and elevation/altitude are still part of the latent variables limnology and geodiversity, respectively, although they should be excluded from those latent variables in model 4 where the "individual influence of elevation and conductivity was tested on species composition and richness."

Answer: We re-run the SEM analysis excluding elevation and conductivity from geodiversity and limnology latent variables respectively. We update SEM output, global metrics of fit and resulting graph. Results and discussion were updated as well, but they are only slightly modified as no major changes were observed.

[Figure]

Biogeosciences Discuss., author comment AC1
https://doi.org/10.5194/bg-2021-298-AC1, 2022

[Figure]

**Reply on RC2 (second revision)**

Laura Anahí Macario-González et al.
* * *
Author comment on " Geodiversity influences limnological characteristics and freshwater ostracode species distributions across broad spatial scales in the northern Neotropics" by Laura Anahí Macario-González et al., Biogeosciences, https://doi.org/10.5194/bg-2021-298-AC1, 2022
* * *
**Reviewer 2.-** We appreciate all the comments and suggestions, they significantly improve the manuscript. Below we provide answers to all comments. We addressed most of the suggestions made by reviewer 2, but in case this was not possible, we provide a detailed explanation.

General comments

Geodiversity. I appreciate the defining of "geodiversity" at Line 38. It seems that the term "geodiversity" is synonymous with "local environmental conditions", a term that is more clear and is used nearly exclusively (at least in the multiple studies that I have seen). It's not clear why "geodiversity" is preferred, and in fact the term is misleading -- especially the "diversity" part of the word (which implies differences among sites, rather than denoting a fixed set of conditions). The authors should at least indicate this synonymy, e.g. near Line 38. Note that this is especially misleading at Line 618. I encourage the authors to consider replacing each "geodiversity" with "environmental conditions" (or a variation thereof) throughout, unless there's a convincing reason that "geodiversity" is indeed the better term.

Answer: The term geodiversity is defined by various authors as follows:

1. The "Encyclopedia of Geomorphology (edited by A. Goudie) Zwolinski states for the term "geodiversity" (p. 417): The most popular definition of geodiversity was put forward by the Australian Natural Heritage Charter (AHC 2002): Geodiversity means the natural range (diversity) of geological (bedrock), geomorphological (landform) and soil features, assemblages, systems and processes. Geodiversity includes evidence of the past life, ecosystems and environments in the history of the earth as well as a range of atmospheric, hydrological and biological processes currently acting on rocks, landforms and soils."

2. Gray, M. (2004: Geodiversity: Valuing and Conserving Abiotic Nature. John Wiley & Sons Ltd., Chichester) presents different definitions of the term geodiversity (see Table 1 on page 10). Often the definitions include materials (similar to our study) such as minerals, rocks, sediments, fossils, soils and water.

3. Schrodt et al. (2018 Journal of Biogeosciences) defines geodiversity as the variety of geological, geomorphological, pedological and hydrological features and processes, although some definitions include atmospheric aspects, too.

We are convinced that in our study the use of the term "geodiversity" is justified, as we integrate at least 5 aspects of the term such as geology (bedrock), mineralogy, sediments, landform (elevation), and hydrological features of landscape. In studies of macro-ecology and biogeography using species distribution, diversity, and richness, a holistic view of the historic causality and currents drivers of observed patterns is gained by using aspects of geodiversity (using the term as it). The term geodiversity is therefore, commonly used together with the term biodiversity, and such integration is strongly recommended by several authors to comprehensive evaluate linkages between them for biogeography, conservation, and climate change studies. "Environmental conditions" may refer to broader aspects of the landscape including atmospheric, soil and water variables and even human disturbances. Below please find attached references of the applied use of the term geodiversity for biodiversity analysis.

Alahuhta, J., Ala-Hulkko, T., Tukiainen, H., Purola, L., Akujärvi, A., Lampinen, R., & Hjort, J. (2018). The role of geodiversity in providing ecosystem services at broad scales. Ecological Indicators, 91, 47– 56. https://doi.org/10.1016/j.ecolind.2018.03.068

Bailey, J. J., Boyd, D. S., & Field, R. (2018). Models of upland species' distributions are improved by accounting for geodiversity. Landscape Ecology, 33(12), 2071–2087. https://doi.org/10.1007/s10980-018-0723-z

Bailey, J. J., Boyd, D. S., Hjort, J., Lavers, C. P., & Field, R. (2017). Modelling native and alien vascular plant species richness: At which scales is geodiversity most relevant? Global Ecology and Biogeography, 26(7), 763–776. https://doi.org/10.1111/geb.12574

Hjort, J., Heikkinen, R. K., & Luoto, M. (2012). Inclusion of explicit measures of geodiversity improve biodiversity models in a boreal landscape. Biodiversity and Conservation, 21(13), 3487–3506. https://doi. org/10.1007/s10531-012-0376-1

Räsänen, A., Kuitunen, M., Hjort, J., Vaso, A., Kuitunen, T., & Lensu, A. (2016). The role of landscape, topography, and geodiversity in explaining vascular plant species richness in a fragmented landscape. Boreal Environment Research, 21, 53–70.

Tukiainen, H., Bailey, J. J., Field, R., Kangas, K., & Hjort, J. (2017). Combining geodiversity with climate and topography to account for threatened species richness. Conservation Biology, 3, 1–37. https://doi.org/10.1111/cobi.12799

All authors again agreed to use the term "geodiversity", as it reflects best the content of our paper and thus fits perfectly – moreover this makes our study even more attractive to a much larger number of scientists from different disciplines.

Sample sizes. Apparently only six individuals (or three?) were identified at some sites, but as many as 60 were at other sites. A better mention of the numbers identified is essential, for example at Line 217. This creates the potential for a substantial error due to sampling effort. For sites with six identified individuals, the percentages are coarse approximations at best. This study is still useful – we all deal with compromised or limited data sets -- but this potential limitation must be addressed in the Discussion.

Answer: We realized that our text is not clear enough to explain our sampling procedure. Sampling was standardized for all lakes to avoid differences in sampling effort. In most lakes, 5 littoral water samples and 5 sediment samples were collected (a total of ten samples). In five large lakes such as Lake Nicaragua, Lake Atitlán, which exceed 100 km$^2$, we collected seven water and seven sediment samples (14 samples). Out of these samples, we sorted individuals and separated them based on their external morphology. The identification of the morphological groups was conducted by dissecting and evaluating taxonomic attributes in at least three individuals. Based on this, we consider that our sampling effort did not bias our diversity index calculations. Now, we clearly explained the sampling procedure in the manuscript.

Specific comments

Comments: Line 156: Please give the reader an idea of how many samples were *typically* collected. If most lakes were represented by two samples (i.e. 6 individuals) but some were represented by ten samples (i.e. 30 individuals), this may affect the apparent distribution of species as well as the calculated richness and Shannon-Weiner diversity indices.

Answer: sampling size was standardized in all lakes to reduce bias related to total and relative abundance, richness and diversity. In almost all lakes five littoral and five sediment samples were collected and analyzed. As lake size matters and may produce over- or underestimation of diversity because of sampling effort. For six large lakes from Central America, seven littoral and seven sediment samples were evaluated. This is explained in detail in lines 156-162.

Line 215: Change "was" to "were".

Answer: done

Line 216: Either transpose "stereomicroscope" and "Leica MZ75", or place "Leica MZ75" in parentheses.

Answer: Following the suggestion of the reviewer we added "Leica MZ75 stereomicroscope"

Line 217: It seems that only 3 individuals were identified in each sample, and up to 10 samples were collected at each lake (Line 156); please report the average number of samples per lake.
Multiple sampling is appropriate and should be repeated here at Line 217. I'm not familiar with ostracod procedures, but three is a very small number. If this is a reasonable number, please indicate and include references. For example, add "Although a sample size of three is
very small, this is consistent with ostracod studies in general (reference, year) in part due to low
ostracod abundances in most sediments worldwide (reference, year)." In addition, see comment for Line 379.

Answer: For species identification, we first sorted ostracodes in all samples (mostly 10 samples per lake) and then, individuals were separated based on their external morphology (morphospecies). For each morphospecies a minimum of three individuals were dissected to identify them taxonomically at species level. Please consider that at each lake we recognize between 2-7 morphospecies. We describe this procedure in detail in the lines 219-225 in the methodology section.

Line 244: Delete "the" and transpose "software" and "Canoco version 5".

Answer: done

Line 248: Please specify the software that includes SEM. It may be part of R, but this isn't mentioned until Line 264, so a mention is needed early in this section.

Answer: We report in line 248 the R package used and then deleted it from line 264.

Line 379: Please report the total number of species.

Answer: We found 70 species, with species richness ranging from 1-9 species and 4 species average per lake. This information is described in line 385.

Line 421: I see the groups of ostracods from NMDS is Figure 5(a), but I would very much like to see the eigenvectors from CCA. A figure similar to Figure 4(a), but based on CCA, is important.

Answer: Following the suggestion of the reviewer, we include CCA eigen vectors and explained variation for the first and second axis to the NMDS graph.

Line 423: We're told that local geology is the ultimate driver, and I accept this as true. I think its effect is delivered through evaporative concentration (PCA shows that ionic concentration varies strongly among these systems). In other words, ionic concentration is the immediate cause; what matters *directly* to ostracods is water conditions, so local geology is an *indirect* factor. A figure with NMDS eigenvectors would be very, very useful for this.

Answer: The PCA shows that ionic concentration is highly variable between lakes and regions in our study area. In highlands, the variability of ionic concentration was mainly associated to evaporation and volcanic influence. This same trend is observed in highlands in Central Mexico and Central America. At regional scale, considering lowlands lakes, the role of evaporation in ionic composition is less evident and the role of bedrock seem to be more relevant. This latter patter is recovered in SEM analysis and therefore discussed in the manuscript, in lines 510-530.

Line 447: Insert comma after "limnology".

Answer: done

Line 460 (Figure 6): I like this figure very much; it's clear yet detailed. I'm somewhat confused,
however: doesn't "geodiversity" *include* limnology, conductivity, and elevation? I'm not sure that "NMDS" is needed in the "Species composition" box; please verify and/or put "NMDS" in
parentheses.

Answer: As previously suggested, geodiversity now is more precisely defined and used in the manuscript. For SEM, geodiversity was defined based on bedrock type and age and elevation. Limnology was used as an independent variable and considered to be integrated by conductivity ionic composition, pH. Please see supplementary material S1 section. This discrimination was done to test the individual influence of geodiversity and limnology in ostracode composition and richness. In one of the models evaluated, we tested the individual influence of elevation and conductivity, as they were recognized as most influential for ostracode distribution in CCA. This model more clearly explains the relationships between species and geological and limnological variables.

The NMDS term included in the box of species composition was put in parenthesis.

Line 467: Replace "drives" with "defines".

Answer: done

Line 506: It's not clear that volcanism created most Central American lakes; in my experience, at least as many were formed by mass movement (in higher elevations). At best, the authors could say that "Volcanism is a common mode of formation".

Answer: we agree with the comment of the reviewer and changed the text accordingly.

Line 547: Line 217 reports that three ostracods were identified per sample, yet Line 547 reports up to nine species per lake.

Answer: We address this issue in the methodology section to avoid confusion. Three individuals make reference to the number of specimens dissected for identification of morphotypes observed in the lakes. We usually observed 1-9 morphotypes per lake.

Line 551: Collection effort can make a huge difference here. If ten samples were collected from one lake but only two from another, we'd expect greater richness in the 10-sample lake. In addition, two samples may be enough to characterize small lakes (*if* sufficient numbers of ostracods are identified in each sample), but this is less true for larger lakes. At least this should be acknowledged here: "While the number of samples per lake varied from X                                                    to                                                    Y"….

Answer: following the comment of the reviewer, we included the number of samples per lake. Please see general comments.

Line 618: transpose "abundant aquatic systems" and "high geodiversity". If "geodiversity" =
"environmental conditions", then this phrase should be "high diversity of environmental conditions" rather than "geodiversity". This is where the "diversity" in "geodiversity" becomes
confusing.

Answer: We modified the text as follows "The northern Neotropics is a region characterized by diverse environmental conditions, abundant aquatic systems, and high biodiversity". This is a general statement, and environmental conditions is preferred over geodiversity, because it makes reference of to the set of atmospheric, soil and water variables.

---

## Author Response (AR3)

Biogeosciences Discuss., author comment AC1
https://doi.org/10.5194/bg-2021-298-AC1, 2022

[Figure]

**Reply on Editor comments**

Laura Anahí Macario-González et al.
* * *
Author comment on "Geodiversity influences limnological conditions and freshwater ostracode species distributions across broad spatial scales in the northern Neotropics" by Laura Anahí Macario-González et al., Biogeosciences, https://doi.org/10.5194/bg-2021-298-AC1, 2022
* * *
Dear Editor,

We appreciate all comments and suggestions made to improve the manuscript. Please find below answers for each of your comments. In case we were unable to follow them, we provide an explanation.

General comments

Comment 1) I strongly side with a reviewer comment about the usage of the word "geodiversity". You kindly provide definitions, which clearly include geodiversity being linked to the RANGE (DIVERSITY) and VARIETY of geological conditions. You never compute nor use any measure of geodiversity. This could be done for a region and meaningfully linked to biodiversity at a larger spatial scale, for instance for limnological subregions, but it´s not the subject of your study. Please, reword the manuscript and title, refer to "geological conditions" or "environmental conditions" instead.

Answer: Definitions provided clarify the coverage of the term geodiversity, the variables that integrate it and its applicability at spatial scale. Our study integrates at least five variables that constitutes geodiversity such as geological units and ages, mineralogy, elevation, and hydrological features of landscape. In addittion, the study area covers a region >1300 km N-S for which we accounted large spatial scale. The suggested terms „geological conditions and environmental conditions" are innapropriate for our database as both are too broad and unspecific. We consider that the usage of the term geodiversity in our study is supported by our data and we retained in the title and through the manuscript.

Comment 2) A similar yet less prominent problem pertains to the usage of the word "limnology", which literally means "the STUDY OF inland waters". You use this word as a hypernym for limnological conditions, defined as including biological variables. However, in fact, you never include any biological variables (a classic would be chlorophyll-a) and instead analyse the relationship to a biological dataset. Here, I simply suggest to use "limnological conditions" (then clearly defined as excluding biological data) or - maybe even better - "water chemical conditions".

Answer: Following the suggestion, we replace the term „limnology" by „ limnological conditions". Please note that limnoclogocal conditions is also used in the title. Definition of

the term is provided in the introductory section of the manuscript (line 73). We retained the biological component of the term as it may not be solely regarded to chlorophyll but to a broad range of biological diversity. Limnology as discipline must be considered as integrative. Please see Limnology - an overview | ScienceDirect Topics.

Comment 3) I believe that the reviewer concern about low species richness being potentially based on low sampling effort and low counts of individuals has been addressed only partly by providing additional information on sampling and abundances. Please, also include a proper discussion of this problem of sampling effort. This seems specifically necessary as there is a lot of text discussing differences to more species-rich systems elsewere that were likely covered with a much higher sampling effort. It´s quite natural that single Lake Ohrid is sampled with more effort than >70 systems in a single study.

Answer: The goal of this study was to take advantage of the high degree of geodiversity in the research area, and to show the linkage between geodiversity and biodiversity using species assemblages of ostracodes as example. We, however, discussed briefly the relative low sampling effort and how temporality may affect species richness.

Comment 4) I have to also side with reviewer concerns about the SEM. First and foremost, I have difficulties to see the added benefit of the SEM to an already long manuscript. The SEM is rather complicated derived with several steps and the procedures can barely be followed. For instance, similar to Reviewer 1, I have difficulties to understand the SEM output and how a metavariable like "limnology" can be included, which consists of several variables yet is then finally presented with only one path coefficient to other variables in the SEM. Similarly, I do not see a clearly justified procedure how an individual variable like conductivity is isolated from the "limnology" dataset and maybe even used twice? Honestly, the simplest way to address this issue would be to just drop the SEM from the manuscript. Alternatively you may provide a better explanation of the procedures and clearly provide the added benefit to the paper.

Answer: SEM analysis was retained, because is the only test in the study able to evaluate relationships between latent variables (variables composed or formulated out from other measured variables) such as geodiversity and limnological conditions and test their statisticall significance. SEM analysis is justified in our study, as the main objective of the manuscript is to evaluate the relationship of the geodiversity and limnological conditions with species composition to infer direct or indirect influences between them. Such analysis can only be adressed using tests able to handle causality assumptions on a multivariable framework, which is the case of SEM. Please see the application of SEM to ecological studies Applications of structural equation modeling (SEM) in ecological studies: an updated review | Ecological Processes | Full Text (springeropen.com)

The reviewers concers were in fact, the variable selection and how they were *a priori* related, they did not question it uses on the analysis. The way how the individual variables were used to fit models evaluated are clearly explained in the main text and in supplementary material. For instance, performance of individual variables were not in any case tested double.

Comment 5) One of two objectives of the paper is the linkage between geology/limnology with ostracod assemblages. I honestly think that this objective has been adressed rather poorly. Also considering the storyline of the paper that is so much based on ordinations. There are several straightforward options to improve the paper in this regard: Consider opposing the maps resulting from the various ordinations - do groupings based on geo data overlap with those based on ostracods? Consider using clusters resulting from geo/limno data as color codes in the ordination based on biological data or even in the PERMANOVA. Last, there is a CCA hidden in the Supplementary that could be graphically improved and presented in the main text. I consider these options as cleaner solutions to address this objective than the SEM. Please consider these suggestions as late editorial ideas to improve the readability and accessibility of your manuscript. You may not follow them.

Answer: All suggestion were tested, one important problem is the interpetation of such graph overlaps, they all must be analyzed based on observation which may imply subjectivity. Another important issue is that geodiversity and limnological conditions are composed on a set of variables. In an ordination is not possible to clearly distinguish the signal of composed variables, beacuse its components are dispersed in the ordinations. The same apply to limnological conditions. Therefore, we cannot infer reationships between variables of interest. The CCA or any other ordination can only support SEM results. In correspondence with the next comment (shortening the manuscript), we decided to exclude the CCA of the manuscript.

Comment 6) Last, please consider shortening the manuscript. Apart from several rather long text sections I can see two clear options: Move table 1 to the SI. And consider computing just one PCA instead of two. I see no justification of computing two PCAs for two subregions anyway. It´s also unclear why the variable selection was done only for one PCA and not for the other. Also, the two PCAs clearly are similar (especially PC1) anyway. Instead of presenting two PCAs you may consider just one, but then present three PCs rather than just two. Also, the contour plots could be based on PC-scores instead of selected variables.

Answer: Table 1 was removed from the manuscript as it shows a synthesis from table S1, which is the entire data set used in the manuscript. In regard of the PCA, lines 199-200, describe how the analyses were performed. Using the cluster results, we construct two databases which where subject to two PCA runs. The first run was done to variable selection. In order to keep the manuscript short, we decided not include 4 PCA graphs, but only second PCA runs. The contour plots were based on variable importance indicated by PC scores as pointed out in the figure caption. Maps show the spatial distribution of variables.

---

## Author Response (AR4)

Biogeosciences Discuss., author comment AC1
https://doi.org/10.5194/bg-2021-298-AC1, 2022

[Figure]

**Reply on Editor comments**

Laura Anahí Macario-González et al.

Author comment on "Geodiversity influences limnological conditions and freshwater ostracode species distributions across broad spatial scales in the northern Neotropics" by Laura Anahí Macario-González et al., Biogeosciences, https://doi.org/10.5194/bg-2021-298-AC1, 2022

Dear Editor,

we appreciate all comments and suggestions made to improve the manuscript. Please find below answers for each of your comments. In case we were unable to follow them, we provide an explanation.

General comments by Associate Editor
I have now read your last submitted manuscript and have to say that I remain seriously confused about your usage of the term "geodiversity". I have no doubt that the aquatic ecosystems you investigated are located in a geologically diverse region. Surely, this geodiversity may have an imprint on the biodiversity at the larger spatial scale of the whole investigated region. The mechanism behind this is likely that geological conditions drive limnological conditions that then shape the composition of communities, here exemplified by ostracod assemblages. So, what I see is the use of the word "geodiversity" for two very different things:

1) The actual range or variety of geological conditions in a given region. This agrees with the definition provided in the very first sentence of the introduction. I think it is also fine to use the word in this sense for headlines, to illustrate general linkages (e.g. the connection of geodiversity and biodiversity) or to interpret larger-scale implications of your findings (e.g. for biodiversity in the region).

2) The (average) geological conditions at a given sampling site. This agrees with the way you compute and use variables that somehow describe geology. There is no single variable in the entire manuscript that actually expresses geodiversity in the sense defined. Also the latent variable in the SEM which is called "geodiversity" can´t actually express the variety of ecological conditions at any site, as it plainly inherits the "state"-nature of the variables it is allowed to be controlled by (lines 248-249). Prominent examples for this usage include the second question which the MS aims to address (line 103), and - more importantly - is exactly aligned with the main conclusions of the manuscript (see, for instance, lines 495-501 or the complete chapter 4.1.).

This "double-use" of the word geodiversity is in the end simply confusing for the reader. It causes statements like "geodiversity...is...constant in [region X]", which I have difficulties to understand. It suggests sloppy usage of the word geodiversity and hard-to-follow statistical analysis. And it finally gives the reader the impression of a "broken promise".

It is my responsibility as an associate editor to assure quality of published manuscripts. In this role, I recommend to reconsider the usage of the word "geodiversity". In your MS, it may be appropriate in some places, yet not so much in others. A "clean" use will benefit the paper. Ultimately, I will leave this decision to the team of authors, however, and will not reject the paper if you decide to just stick to how you used it.

Answer by authors:

In regard of the term geodiversity, we consider that the main point of disagreement between you and our multidisciplinary team, is how this term is interpreted. We consider geodiversity as an integrative concept, and, in lines 39-45, its components are described. We used the view of Zarnetske et al. (2019) who describe different approaches to evaluate geodiversity in a region of interest. Please consider that using a single variable (e.g. elevation) or a broad range of geodiversity-related variables it is possible to study the relationships between biodiversity and geodiversity.

We approach geodiversity based on at least nine geological and mineralogical variables. This is in counterpart of what you write "no single variable in the entire manuscript actually express geodiversity in the term defined". We clarify our initial concept to focus on an integrative approach, see line 103-106. Furthermore, in the introduction, we now explain how geodiversity can be studied based on measurements of its components.

The SEM analysis does not intend to express the variety of ecological conditions of a single site, but rather explains causal relationships between tested variables without considering a geographical extent (e.g., influence of limnological conditions on biodiversity). In this case, following our initial and integrative concept, the term geodiversity in SEM is constructed using geological and mineralogical variables (lines 247-249) and different variable combinations were used in the models tested. Therefore, we consider that the term geodiversity in SEM agrees with the definition provided in the introduction.

Considering that our team favors the usage of the term geodiversity, and that two anonymous reviewers were satisfied with the explanation we provided for geodiversity, we decided to keep the term throughout the manuscript. We hope that our revision now provides a clean use of the term geodiversity to avoid misinterpretations.